

# The Community Fire Behavior Model for coupled fire-atmosphere modeling: Implementation in the Unified Forecast System

Pedro A. Jiménez y Munñoz[1], Maria Frediani[1], Masih Eghdami[1], Daniel Rosen[1], Michael Kavulich[1], and Timothy W. Juliano[1]

[1]NSF National Center for Atmospheric Research

**Correspondence:** Pedro A. Jiménez y Muñoz (jimenez@ucar.edu)

**Abstract.** There is an increasing need for simulating the evolution of wildland fires. The realism of the simulation increases by accounting for feedbacks between the fire and the atmosphere. These coupled models combine a fire behavior model with a regional numerical weather prediction model and have been used for fire research during the last decades. This is the case, for instance, of the state-of-the-art Weather Research and Forecasting model with fire extensions (WRF-Fire). Typically, the coupling includes specific code for the particular models being coupled such as interpolation procedures to pass variables from the atmospheric grid to the fire grid, and vice versa. However, having a fire modeling framework that can be coupled to different atmospheric models is advantageous to foster collaborations and joint developments. With this aim, we have created, for the first time, a fire behavior model that can be connected to other atmospheric models without the need of developing specific low-level procedures for the particular atmospheric model being used. The fire behavior model, referred to as the Community Fire Behavior model (CFBM), closely follows WRF-Fire version 4.3.3 methods in its version 0.2.0, and makes use of the Earth System Modeling Framework library to communicate information between the fire and the atmosphere. The CFBM can be also run offline using an existing WRF simulation in what we refer to as the standalone model. Herein we describe the fire modeling framework and its implementation in the Unified Forecast System (UFS). Simulations of the Cameron Peak Fire performed with UFS and WRF-Fire are presented to verify our implementation. Results from both models, as well as with the standalone version, are consistent indicating a proper development of the CFBM and its coupling to the UFS-Atmosphere. These results, and the possibility of using the fire behavior model with other atmospheric models, provide an attractive collaborative framework to further improve the realism of the model in order to meet the growing demand for accurate wildland fire simulations.

## 1 Introduction

Many of the hazards posed by wildland fires can be mitigated with accurate predictions of the fire evolution. These predictions can be based on models of a wide complexity (Sullivan, 2009a, b, c) ranging from empirical models that require just the surface winds to drive the fire evolution to combustion models that resolve relevant atmospheric chemistry. For real-time applications coupled fire-atmosphere models provide a balance between the realism of the physical processes represented and the computational resources required to run the model. For instance, Jiménez et al. (2018) showed the potential for real-time applications





of coupled fire-atmosphere models using simulations performed at 111/27.75 m grid spacing for the atmospheric/fire compu-
tational mesh. These models consist of a fire behavior model coupled with a numerical weather prediction (NWP) model, the
atmospheric component, to explicitly account for fire-atmosphere interactions.

A number of coupled fire-atmosphere models are in use nowadays (Peace et al., 2020). For example, the Weather Research
and Forecasting (WRF, Skamarock et al., 2021) model includes a state-of-the-art fire behavior model known as WRF-Fire
(Mandel et al., 2011; Coen et al., 2013). Essentially, the atmosphere informs the winds, surface roughness, and other surface
variables to the fire model wherein these variables are used to drive the fire evolution. The fire evolution takes place on a refined
grid which allows the user to better represent elevation and fuels, and the propagation of the fire. The fuels in the corresponding
grid cells are ignited at the pass of the fire front which is tracked with the level set method (Mandel et al., 2011; Muñoz-Esparza
et al., 2018). The ignited grid cells release heat and moisture, and the fire component adds the vertical flux divergences back
into the atmosphere as temperature and moisture tendencies. The smoke produced by the combustion of the fuels is transferred
to the atmosphere, where it is advected and diffused as a tracer. Other fire-atmosphere coupled models follow similar strategies
to provide feedback to the atmosphere (e.g. Filippi et al., 2011; Coen, 2013; Dahl et al., 2015).

A non-negligible challenge in the fire-atmosphere coupling is the necessity of interpolating variables from the atmospheric
grid to the fire grid and vice versa. This is especially the case when coupling fire behavior models with non-structured atmo-
spheric grids or grids with unequal distance between grid points. To address this limitation, grid remapping or regridding, is
done using software especially developed to couple models with different grids. For example, one can use the Earth System
Modeling Framework (ESMF, Balaji et al., 2023), which is a library that facilitates interoperability between model compo-
nents, to couple the fire and atmosphere components as it has been done to couple other Earth system components (e.g., Sun
et al., 2019; Bauer et al., 2021). Following ESMF standards, this requires implementing the model to be coupled as a national
unified operational prediction capability (NUOPC). The NUOPCs are independent models of Earth system components with
procedures to initialize and advance the component. Several NUOPCs can be run jointly in a coupled mode building an Earth
system application also using the ESMF library to coordinate the evolution of the components. However, we are not aware of
any fire behavior model available to couple with atmospheric models in such a way.

Herein we present, for the first time, a fire behavior model implemented as a NUOPC to facilitate its coupling with other
atmospheric models. The fire behavior NUOPC is publicly available and we refer to it as the Community Fire Behavior model
(CFBM). The model is based on WRF-Fire version 4.3.3 procedures and can be coupled to the atmospheric component via
the ESMF infrastructure. We illustrate its coupling with the atmospheric component of the Unified Forecast System (UFS-
Atmosphere), and compare results between UFS-Fire and WRF-Fire to verify the implementation. During the process of
building the fire model, we created a standalone version of the code that can be run using the output of a previous WRF
simulation (offline coupling). This standalone version of the code runs faster because there is no need to run the atmospheric
component which is more computationally demanding than the fire. Results from the standalone fire code are also compared
with WRF-Fire and UFS. Our results highlight the potential of UFS to simulate the evolution of wildland fires, allowing UFS
users for the first time to explicitly model fire-weather interactions to conduct fundamental and applied fire research. The
CFBM can be coupled to other atmospheric models which has the potential to foster joint developments with the ultimate goal





of advancing fundamental understanding of wildland fires and their impacts on the Earth system, and, from a more applied point of view, contribute to minimizing the deleterious impacts of wildland fires.

The manuscript is organized as follows. The next section describes the fire behavior model whereas Section 3 describes its implementation in UFS. The comparison of fire simulations with UFS, the standalone version of the code, and WRF-Fire for a wildland fire event, the Cameron Peak Fire, is shown in Section 4. Finally, the conclusions are presented in Section 5.

## 2 The Community Fire Behavior Model


The structure of the code is shown in Figure 1. Each of the major portions of the code is described in the following sections that focus on the fire behavior model (Section 2.1), the ESMF extensions to create the NUOPC (Section 2.2), and the automatic compilation and testing (Section 2.3).

**Community Fire Behavior NUOPC**

| Standalone code | ESMF code | Compilation and testing |
|---|---|---|
| shared | nuopc | **CMake** |
| io | | cmake |
| state | | env |
| physics | | **CMakeLists.txt** |
| driver | | tests |
| | | **compile.sh** |

**Figure 1.** Diagram illustrating the organization of the CFBM code. Each box corresponds with a directory in the code.

### 2.1 The fire behavior model

The fire behavior model is based on WRF-Fire version 4.3.3. In WRF, the user defines the atmospheric domain and a refinement ratio to have finer horizontal grid spacing on the fire grid. Hence, the atmospheric and the fire grids cover the same region, but the fire grid has finer grid spacing. It is also required to provide a fuel dataset, based on the 13-fuel categories from Anderson (1982) or the 40-fuel categories of Scott and Burgan (2005), and an elevation dataset to interpolate the data onto the fire grid.





The elevation dataset is used to calculate the slope in the south-north and west-east directions, which is then used by the fire
model to propagate the fire. Different elevation datasets can be used for the atmosphere and the fire given their different grid
spacing. The domain configuration and the interpolation of the static datasets are performed by the Geogrid program which is
part of the WRF Pre-Processing System (WPS). In the CFBM, we also rely on the Geogrid program to define the fire grid and
interpolate the static datasets, but, since there is no atmosphere component, only the fuels, and elevation (including slopes) in
the fire grid are used.

Ignitions can be performed in two ways. One option is to set up the ignition parameters in the fire namelist. The ignition
can be a point, a circle, or a line ignition, and the ignition time is set up as seconds since the beginning of the simulation. The
other option is to initialize the fire perimeter from observations (e.g. fire mappings from aircrafts). The observations are used
to initialize the level set function which defines the location of the fire front. The level set function is a signed distance function
from the fire front, positive (negative) outside (inside) the fire perimeter (Mallet et al., 2009). Hence, at the fire perimeter the
level set function is zero. The user is required to generate and add this initial level set function to the file generated by the WPS
Geogrid program. The ignition time from the perimeter is also defined in the fire namelist as seconds from the beginning of the
simulation.

Once a fire has been ignited, the fire front is propagated based on a parameterization of the fire rate of spread ($R$). Currently,
only ground fires are considered and $R$ is parameterized based on Rothermel (1972). $R$ depends on the fuel characteristics, the
surface winds, and the terrain slope using

$$R = R_0(1 + \phi_w + \phi_s) \tag{1}$$

where $R_0$ is the rate of spread over flat terrain for a case with no wind; $\phi_w$ is the wind correction, and $\phi_s$ is the slope correction.
$R_0$ depends on only the fuel characteristics. Some of the fuel characteristics are constant and others are a function of the fuel
type defined by the 13 Anderson categories. The use of the more refined fuel models of Scott and Burgan (2005) is supported
with a mapping of its 40 categories into the 13 categories of Anderson (1982). The fuel-dependent variables (Table 1) are
the fuel load of the surface fuel, the fuel bed depth, the surface area to volume ratio, the characteristic time scale for burnout
rates, and the fuel moisture content (FMC, the mass of water per unit mass of dry fuel) of extinction. The FMC can be either
set to constant (8% by default), or estimated dynamically using a FMC model (Mandel et al., 2014). If the FMC model is
used, the mass of the surface fuels for 1 h, 10 h, 100 h, and 1000 h dead fuels and live fuels (Table 2) for each fuel type is
also used. The constant characteristics are the ovendry fuel density ($32\,\mathrm{lb\,ft}^{-3}$), the fuel particle total mineral content (5.55%),
and the fuel particle effective mineral content (1.0%). The fuel information is used to calculate $R$ for the case of no wind and
flat terrain ($R_0$). In the presence of winds, the wind correction, a function of the fuel characteristics and the surface wind,
is also needed ($\phi_w$). The surface wind responsible for the fire propagation are interpolations of the three-dimensional winds
from the atmospheric component (see next section) to a height specified by the user. Two interpolation options are available:
interpolation from two adjacent model layers to the target height, and interpolation by a given height to the target height based
on the logarithmic wind profile. This last option was implemented to use upper levels winds less affected by the fire since
Rothermel's parameterization was designed to use ambient winds. Similar to the wind correction term, the effects of the terrain





**Table 1.** Some characteristics of the Anderson's 13 surface fuel models used in the CFBM

| Fuel model name | Fuel load | Fuel bed depth | Moisture of extinction | Surface area to volume ratio | burn time |
|---|---|---|---|---|---|
| | [$\mathrm{kg\,m^{-2}}$] | [m] | [−] | [$feet^{-1}$] | [s] |
| 1. Short grass | 0.166 | 0.305 | 0.12 | 3500 | 7 |
| 2. Timber (grass and understory) | 0.896 | 0.305 | 0.15 | 2784 | 7 |
| 3. Tall grass | 0.674 | 0.762 | 0.25 | 1500 | 7 |
| 4. Chaparral | 3.591 | 1.829 | 0.20 | 1739 | 180 |
| 5. Bush | 0.784 | 0.610 | 0.20 | 1683 | 100 |
| 6. Dominant bush, hardwood slash | 1.344 | 0.762 | 0.25 | 1564 | 100 |
| 7. Southern rough | 1.091 | 0.762 | 0.40 | 1562 | 100 |
| 8. Closed timber litter | 1.120 | 0.061 | 0.30 | 1889 | 900 |
| 9. Hardwood litter | 0.780 | 0.061 | 0.25 | 2484 | 900 |
| 10. Timber (litter and understory) | 2.692 | 0.305 | 0.25 | 1764 | 900 |
| 11. Light logging slash | 2.582 | 0.305 | 0.15 | 1182 | 900 |
| 12. Medium logging slash | 7.749 | 0.701 | 0.20 | 1145 | 900 |
| 13. Heavy logging slash | 13.024 | 0.914 | 0.25 | 1159 | 900 |

slope are also accounted for by an additional term ($\phi_s$). This term depends on fuel properties and the terrain slope. The only exception to the Rothermel rate of spread parameterization is the chaparral fuel model that uses a function that only depends on the wind speed (Clark et al., 2004). In this way, the fuel characteristics, winds, and terrain slope determine the fire rate of spread.

The rate of spread is used to propagate the fire by advancing the level set function (Mandel et al., 2011; Muñoz-Esparza et al., 2018). This is accomplished by numerically solving the level set equation:

$$\frac{\partial \varphi}{\partial t} + R(\|\nabla \varphi\| - \epsilon \Delta \varphi) = 0 \tag{2}$$

where $\varphi$ is the level set function, $\nabla \varphi = (\frac{\partial \varphi}{\partial x}, \frac{\partial \varphi}{\partial y})$, and $\epsilon \Delta \varphi$ is an artificial viscosity stabilizer with $\Delta \varphi = \Delta x \frac{\partial^2 \varphi}{\partial x^2} + \Delta y \frac{\partial^2 \varphi}{\partial y^2}$, and $\epsilon$ the artificial viscosity constant. Here, $R$ is the rate of spread of the fire normal to the fire front which is calculated using the wind speed and wind slope normal to the fire front in Rothermel's rate of spread parameterization (Eq. 1). The normal unit vector to the fire front needed for the projections is $\frac{\nabla \varphi}{\|\nabla \varphi\|}$. The level set equation is solved using a third order explicit Runge-Kutta scheme for the time integration. For the spatial derivatives, several methods are available. By default, the spatial derivatives are solved using the fifth-order weighted essentially nonoscillatory (WENO5) method around the fire front and a first-order essentially nonoscillatory (ENO1) method elsewhere. This allows for an accurate solution of the level set function near the fire front, and, at the same time, minimize the computational cost elsewhere when the value of the level set function is not relevant. The term representing the artificial viscosity requires two values of the viscosity, one near the fire front and





**Table 2.** Fuel loads for the 1 h, 10 h, 100 h and live fuels [kg m$^{-2}$] for the Anderson's 13 surface fuel models . The 1000 h load is set to zero.

| Fuel model name | 1 h | 10 h | 100 h | live |
|---|---|---|---|---|
| 1. Short grass | 0.74 | 0.00 | 0.00 | 0.00 |
| 2. Timber (grass and understory) | 2.00 | 1.00 | 0.50 | 0.50 |
| 3. Tall grass | 3.01 | 0.00 | 0.00 | 0.00 |
| 4. Chaparral | 5.01 | 4.01 | 2.00 | 5.01 |
| 5. Bush | 1.00 | 0.50 | 0.00 | 2.00 |
| 6. Dominant bush, hardwood slash | 1.50 | 2.50 | 2.00 | 0.00 |
| 7. Southern rough | 1.13 | 1.87 | 1.50 | 0.37 |
| 8. Closed timber litter | 1.50 | 1.00 | 2.50 | 0.00 |
| 9. Hardwood litter | 2.92 | 0.41 | 0.15 | 0.00 |
| 10. Timber (litter and understory) | 3.01 | 2.00 | 5.01 | 2.00 |
| 11. Light logging slash | 1.50 | 4.51 | 5.51 | 0.00 |
| 12. Medium logging slash | 4.01 | 14.03 | 16.53 | 0.00 |
| 13. Heavy logging slash | 7.01 | 23.04 | 28.05 | 0.00 |

the other valid elsewhere (by default the value of both viscosities is set to 0.4). The motivation for this is to allow for an

accurate solution of the level set equation near the fire front, and facilitate stability of the numerical method. However, as the fire propagates, the level set function needs to be reinitialized to maintain its properties (signed distance function from the fire front). The reinitializations require solving the equation

$$\frac{\partial \varphi}{\partial \tau} + S(\varphi_0)(\|\nabla \varphi\| - 1) = 0 \tag{3}$$

where $S(\varphi_0) = \varphi_0(\varphi_0^2 + \Delta x^2)^{-1/2}$, $\varphi_0$ is the level set function after solving the level set equation (Eq. 2), and $\tau$ is a pseudo

time (Sussman et al., 1994; Muñoz-Esparza et al., 2018). Considering the importance of reinitializing the properties of the level set function, reinitializations are activated by default. The reinitialization equation is also solved using a third order explicit Runge-Kutta scheme for the time integration and WENO5/ENO1 for the spatial derivatives. Solving the reinitialization equation requires iteration. One iteration is set as default base on our previous work (Muñoz-Esparza et al., 2018). Hence, the level set equation (Eq. 2) and its reinitialization equation (Eq. 3) are solved every time step in order to advance the level set

function that tracks the evolution of the fire front.

After advancing the level set function, the fire can propagate into adjacent grid cells which are ignited at the pass of the fire front. Actually, each grid cell is subdivided into four parts and the level set function is interpolated to these refined grid to better track the propagation as well as the estimation of the burnt area and the fuel consumption by combustion (Mandel et al., 2011). The fuel available to burn in a given grid cell depends on the fuel type (Table 1). The time it takes to burn the fuel is

approximated with an exponentially decaying function (Clark et al., 2004; Mandel et al., 2011; Coen et al., 2013) that requires





a characteristic time, the burn time in Table 1, that is different for each fuel type. The burn time produces a decrease of 40% in the fuel amount in 10 min for a value of 1000 s, and it is translated to the e-folding time by dividing the burn time by 0.85 (Mandel et al., 2011). The combustion process generates heat and moisture fluxes that are parameterized. The heat release is represented with the following equation

$$SH = \Delta m f_{dry\_fuel} h_c \tag{4}$$

where $SH$ is the kinematic sensible heat flux released by the ground fire [J m$^{-2}$ s$^{-1}$]; $\Delta m$ is the fuel mass burned [kg m$^{-2}$ s$^{-1}$], $f_{dry\_fuel}$ is the fraction of dry fuel mass per fuel mass defined in terms of the FMC following $f_{dry\_fuel} = \frac{1}{1+FMC}$; and $h_c$ is the heat of combustion for dry cellulose (17.433 MJ kg$^{-1}$). The kinematic latent heat flux ($LH$ [J m$^{-2}$ s$^{-1}$]) from the ground fire is

$$LH = \Delta m(f_{water} + f_{water\_in\_cellulose} f_{dry\_fuel})L_v \tag{5}$$

where $f_{water}$ is the fraction of water in the fuel mass ($f_{water} = \frac{FMC}{1+FMC}$); $f_{water\_in\_cellulose}$ is the fraction of water in cellulose released during combustion (56%); and $L_v$ is the latent heat of vaporization of water (2.5 MJ kg$^{-1}$). In addition to the heat and mositure fluxes, a fraction of the fuel burnt during the combustion, 2% by default, is released as smoke.

The procedures of the CFBM described above closely follow WRF-Fire methods, but important modifications have been introduced in the code. Using the same methods is desirable in the version 0.2.0 of the CFBM herein presented to ensure proper implementation as will be shown below. We have reorganized, modified and added substantials portions of new code to create a standalone fire model, with build in tests, and to clarify the logic of what is being done. Additionally, the code is more orthogonal in order to facilitate maintenance and future extensions. For the same reasons, we have added more derived types with their own type-bound procedures. For example, the most important physical processes that are parameterized have an abstract derived type with deferred procedures, and the derived type is extended with a particular parameterization of the process. Furthermore, we have reformatted procedures, to have a consistent format, and self-documented most of the code to facilitate understanding of what is being done. Some procedures had arguments that were passed but not used, and these arguments have been removed. We also have incorporated the implicit none statement in all procedures, and, when possible, functions are declared as pure. A substantial number of consistency checks and prints through the code have been removed to improve readability and performance. The namelist options controling the fire evolution have been simplified keeping only a reduced set. In addition, all the atmospheric dependencies have been removed from the fire code. This includes removing procedures to interpolate atmospheric variables from the WRF grid to the WRF-Fire grid. The calculation of the fire grid latitudes and longitudes also used these procedures to interpolate the latitude and longitude of the atmospheric grid. We now use an improved approach to determine the geolocation of the fire grid points based on the map projection information. The code is compliant with the Fortran 2008 standard.

The CFBM can be driven by an existing WRF simulation in an offline manner in what we refer to as the standalone model. The standalone model was implemented to ensure consistency with WRF-Fire, and thus minimize the number of issues during the developments. With this aim, we generated automatic tests to check the output of a short run of the standalone model against





the WRF-Fire solution. The standalone model does not need the fire grid to match the WRF domain. The only requirements are

the fire grid included within the WRF domain, and the WRF simulation to have certain variables available. These variables are the three-dimensional winds (U and V variables in WRF), the geopotential height (PH and PHB), the roughness length (ZNT), the 2 m temperature (T2), the 2 m water vapor mixing ratio (Q2), the precipitation (RAINC and RAINNC), and the surface pressure (PSFC). The surface pressure, precipitation, 2 m water mixing ratio, and 2 m temperature variables are only used if the FMC model is activated. In addition, the WRF projection must be Lambert-Conformal in CFBM version 0.2.0. The frequency

to update the atmospheric state is controlled by a namelist setting. Although the standalone model was implemented to ensure consistency between WRF and the CFBM, the model is also useful for efficiently testing sensitivities in model parameters and methods. This offline coupling can be extended to other atmospheric models having the basic atmospheric variables described above. The only external library required to compile the standalone fire code are the C and Fortran NetCDF libraries that are used for reading and writing procedures.

The left portion of Figure 1 shows the directories with the fire code. The shared directory contains modules to define the constants of the model, manage dates and times, and handle Lambert-Conformal projections used to initialize the latitudes/-longitudes of the fire grid from the Geogrid file (and by the standalone model to interpolate from the WRF grid to the fire grid). It also contains abstract derived types with deferred procedures for fuels, the rate of spread parameterization, and the FMC model. These types are currently extended with WRF-Fire approaches, but they can be extended with other methods in

order to have different parameterization of these processes. The io directory contains modules to write standard output/errors, to read the Geogrid file, the fire namelist, and WRF variables in the standalone mode. To facilitate the reading of Geogrid and WRF data, the io directory also includes a module with generic procedures to extract information from NetCDF files. The state directory contains a module to define state variables and methods to initialize variables, update atmospheric variables in the standalone mode, and save the state variables in a NetCDF file. There is also a module to divide the domain into tiles for the

upcoming OpenMP parallelization, and another module with a derived type to handle ignition line data and methods to ignite the prescribed fire lines. The fire code is located in the physics directory. This code includes modules with the fire driver, level set procedures, fire physics procedures, and the extension of the abstract types for 1) fuel type following Anderson (1982); 2) the fire rate of spread type following Rothermel (1972); 3) and the dynamic FMC model (Mandel et al., 2014). Finally, the driver directory has the standalone program, and modules to initialize and advance the model that are used by the standalone

program or the ESMF library.

When two-way coupled to an atmospheric model, the ESMF library is also required in order to pass the kinematic fire fluxes (Eqs. 4 and 5) and fire emissions to the atmosphere to account for fire-atmosphere interactions. The coupling framework is described in the following section.

## 2.2 The Community Fire Behavior NUOPC

In order to couple the CFBM to an atmospheric model using the ESMF library, the CFBM is available as a NUOPC. To this end, there is a fire initialization subroutine, a subroutine that advances the fire state one time step, and a file called the NUOPC cap used to communicate with other NUOPC components. The cap calls the fire initialization and advance subroutines, performs





other model initialization tasks, and defines the variables available to import/export by the fire model. The variables that are mandatory to import are the 3 dimensional winds and geopotential height, the roughness length, the 2 m temperature, the 2 m

water vapor mixing ratio, the surface pressure, and precipitation. Precipitation can be passed either as a precipitation rate or as accumulated precipitation since the beginning of the simulation. The variables that can be exported are the kinematic sensible heat flux and the kinematic latent heat flux released from the fire (Eqs. 4 and 5), and the fire smoke emissions. In addition to defining the variables that are available to connect to other NUOPCs, the fire NUOPC cap also defines the fire grid during the initialization. This allows for automatic regridding of the imported/exported variables at run time. In the fire NUOPC cap, we

also perform the vertical interpolation of the 3-dimensional winds into a 2-dimensional array with the winds at the target height above ground level to propagate the fire.

The middle portion of Figure 1 shows the directory providing the ESMF functionality. This is the nuopc directory which contains the fire NUOPC cap. The directory also has files to create an Earth system model (fire driven by an atmospheric component) using ESMF or the Earth System modeling eXecutable (ESMX) for testing purposes (see next Section 2.3). The

ESMX are recent extensions of the ESMF library. Using the ESMX extension reduces redundant code to build the Earth system application, and thus allows for faster developments and easier maintenance. Our fire NUOPC has been an early adopter of this technology. The Earth system model created either with ESMF or ESMX uses the fire NUOPC and a WRF-data NUOPC. The WRF-data NUOPC only has the cap since the WRF-data NUOPC just reads variables from the output of an existing WRF simulation. The cap defines the WRF-data grid and imports the variables that the fire NUOPC needs. This is a one-way

coupling from the atmosphere to the fire and therefore behaves as the standalone model described in the previous section. The main difference is that this offline coupling requires ESMF whereas the standalone code described before does not. The regridding from the atmospheric grid to the fire grid is performed by the ESMF library. The diagram shown in Figure 2 summarizes the coupling strategy between the WRF-data NUOPC and the CFBM NUOPC. The WRF-data NUOPC is used by our automatic tests that quickly check the compliance of the ESMF code.

## 2.3 Compilation and testing

The organization of the code related to compilation and testing is shown in the right portion of the Figure 1. The automatic compilation of the code is implemented with Cmake. A bash script, compile.sh, is used to compile the code and test it if instructed. The test is designed to minimize the number of potential issues introduced into the code. The test runs a series of short simulations of the fire evolution and compares results against the known solution. The test can check the standalone

model build, and the NUOPC build with either ESMF or the ESMX strategies. The ESMF and the ESMX coupling methods rely on the WRF-data coupling described above to perform the test. The github repository of the CFBM also runs the test every time there is a new code development pushed to the repository to minimize the introduction of bugs in the code.

The only necessary libraries to compile and test the code are NetCDF and ESMF (the latter one only for the NUOPC builds). The code can also be built with MPI libraries. Parallelization is not supported in version 0.2.0, but MPI compilation is included

in order to couple CFBM with UFS that uses MPI-based compilation. Cmake is able to find these three libraries in the user environment, but to facilitate compiling in common environments, the env directory contains information to load modules in





**Figure 2.** Diagram illustrating the WRF-data NUOPC and the CFBM NUOPC with the variables that exported (imported) by WRF-data (Fire behavior). The variables in gray are only used if the fuel moisture content model is activated to simulate the moisture content dynamically.

particular systems. At the time of writing, the only computer in env is Derecho, a high performance computer that is part of the NSF NCAR-Wyoming Supercomputer Center.

## 3   Coupling the Community Fire Behavior model to the UFS

The UFS has its components implemented as NUOPCs (e.g., atmosphere, ocean, land, etc.) to facilitate their coupling. Hence, we have coupled the CFBM NUOPC to the UFS-Atmosphere NUOPC using the existing infrastructure.

The coupling diagram illustrating the combined UFS-Atmosphere and the CFBM is shown in Figure 3. The UFS has other NUOPCs that are not illustrated here since it is possible to just run the UFS-Atmosphere together with the fire component. The variables imported by the fire NUOPC are similar to the ones imported for the case of the WRF-data NUOPC (Fig. 2), but the

accumulated precipitation has been replaced by the precipitation rate. The main difference is the feedback provided by the fire NUOPC to the atmosphere via sensible and latent heat fluxes released by the fire. This allows for two-way coupling between the two components if desired. In practice, the user decides if the coupling is one way (information from the atmosphere to the fire only) or two ways (adding the fire feedback to the atmosphere component) via the fire namelist. Smoke emissions as a result of burning the surface fuels by the fire model are also passed to the atmosphere.





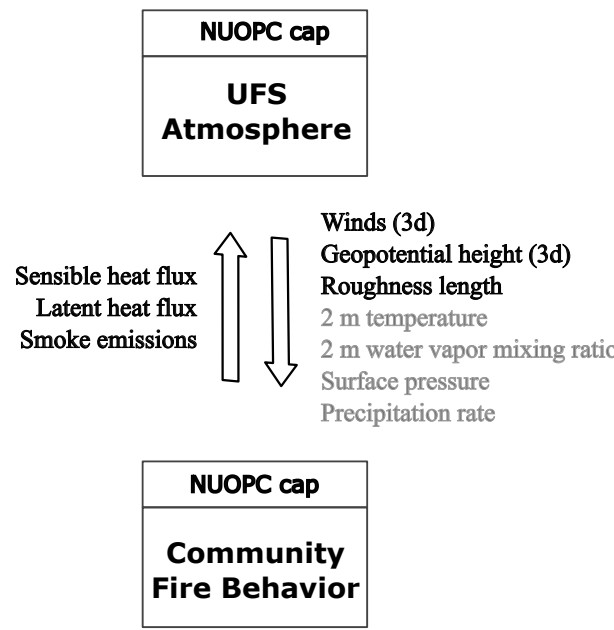

**Figure 3.** Diagram illustrating the UFS-Atmosphere NUOPC and the CFBM NUOPC with the variables that are exported (imported) by the fire behavior (UFS) on the right, and the variables that are exported (imported) by UFS (Fire behavior) on the right. The variables in gray are only used if the fuel moisture content model is activated to simulate the moisture content dynamically.

In UFS, the physical processes, or parameterizations, are represented using the Common Community Physics Package (CCPP, Heinzeller et al., 2023). It is inside CCPP wherein the kinematic fluxes from the fire (Eqs. 4 and 5) are used to update the temperature and moisture tendencies. This is done in the fluxes wrapper. Originally, the wrapper combined surface fluxes from the land, ocean, and ice, but we extended the functionality to include the fire fluxes. The kinematic flux divergences are added to the temperature and moisture tendencies of the first atmospheric layer. The smoke emissions are added to a tracer to

represent the smoke transport and dispersion associated with atmospheric dynamics. A more comprehensive coupling of the smoke with radiation and cloud microphysics is left for future work.

To facilitate running coupled fire-atmosphere simulations with UFS, we have incorporated the fire behavior model into the UFS Short-Range Weather App (SRW App). This is the application that is used to run UFS-Atmosphere in a regional, limited-area configuration, with a workflow that conveniently links pre-processing, running, and post-processing of the model,

with a single configuration file ensuring all components are linked and run together in a self-consistent way. This contribution is a mutually beneficial relationship: the CFBM gets a robust, well-supported workflow for running experiments coupled to an atmospheric model on a wide range of compute platforms, and the UFS community receives new capabilities for fire prediction experiments within the existing, familiar and well-documented framework of the SRW App. With this new capability contributed to the publicly available and supported SRW App, one can perform fire-atmosphere simulations without altering





the steps required to run a conventional atmospheric simulation, using a workflow that allows for easy modification of domains, dates, and other configurable options of the atmospheric component in a way that is consistent with the fire component.

## 4   Model inter-comparison: testing the Community Fire Behavior NUOPC and its coupling with UFS

The version 0.2.0 of the CFBM herein presented closely follows WRF-Fire procedures in order to compare results between the CFBM NUOPC implemented in UFS and WRF-Fire results. This allows us to ensure a proper implementation in order

to have a starting point to introduce further developments to improve the realism of the fire simulations. In order to test the implementation, we have selected a wildland fire over complex terrain in Colorado, U.S., the Cameron Peak Fire, and configured UFS and the CFBM similar to the WRF-Fire configuration. Identical results are not expected considering the different dynamical cores and parameterizations, but the fire evolution should respond to the models' respective atmospheres consistently.

The Cameron Peak Fire started on 13 August 2020 at approximately 20 UTC (2 PM MDT) on the Arapaho and Roosevelt National Forests, west of Chambers Lake in northern Colorado. The fire started East of Cameron Peak, on a hot, dry, and windy day with gusts reaching 71 miles per hour. The fire spread quickly through the mountainous terrain and beetle-killed trees in the region. The fire was contained on 5 December 2020 with an estimated area of around 209,000 acres, representing the largest wildfire perimeter in Colorado's history.

### 4.1   Experimental set up

In this work, we focus on approximately the first 28 h of the fire evolution to minimize the effects of human intervention. Three fire perimeters are available during this period from the Colorado Wildfire Information Management System (CO-WIMS) database. The first perimeter, hereafter referred to as Perimeter-1, corresponds to 2020 August 13 2342 UTC (5:42 PM MDT), the second perimeter, Perimeter-2, to 2020 August 14 1554 UTC (09:54 AM MDT), and the last perimeter, Perimeter-3, to

August 14 2216 UTC (04:16 PM MDT).

A total of 11 fire simulations were performed using either WRF version 4.3.3, UFS, or the standalone CFBM version 0.2.0 (Table 3). A few minor changes were introduced to WRF version 4.3.3. First, we update the atmospheric roughness length every time step since it varies with the simulation time. Second, we decreased the pseudo time for reinitializations by a factor of 100 to avoid numerical instabilities. And third, we updated the VEGPARM.TBL to correct for a bug in the table. Besides the

atmospheric models, the experiments differ in the type of ignition (point ignition or from a perimeter) and the fire-atmosphere coupling strategy (one-way or two-way).

In the point ignition experiments, we simulate the initial 28 hours of the fire evolution, corresponding to a 30-hour model simulation period from 2020 August 13 18 UTC to 2020 August 15 00 UTC. The ignition point was set to 40.609 degrees latitude and -105.879 degrees longitude, with 250 m radius, and it ignited from 6480 to 7000 s after initialization. The first

three experiments in this set, UFS-1way, WRF-1way, and CFB-1way are configured with one-way feedback, meaning the atmosphere affects the fire but the atmosphere does not respond to the fire (i.e., uncoupled). This is the simplest configuration





**Table 3.** Description of the simulations performed. The experiments labelled with CFB use the CFBM withe atmospheric information from the equivalent one-way WRF experiment.

| Experiment | Coupling | Ignition Type | Ignition Time [UTC] | Atmosphere initialization time [UTC] | Fire wind height [m] |
|---|---|---|---|---|---|
| UFS-1way | one-way | point | 2020-08-13 19:48 | 2020-08-13 18:00 | 5 |
| WRF-1way | one-way | point | 2020-08-13 19:48 | 2020-08-13 18:00 | 2.5 |
| CFB-1way | one-way | point | 2020-08-13 19:48 | 2020-08-13 18:00 | 2.5 |
| UFS-2way | two-way | point | 2020-08-13 19:48 | 2020-08-13 18:00 | 5 |
| WRF-2way | two-way | point | 2020-08-13 19:48 | 2020-08-13 18:00 | 2.5 |
| UFS-P1 | one-way | Perimeter-1 | 2020-08-13 23:42 | 2020-08-13 21:00 | 5 |
| UFS-P2 | one-way | Perimeter-2 | 2020-08-14 15:54 | 2020-08-14 15:00 | 5 |
| WRF-P1 | one-way | Perimeter-1 | 2020-08-13 23:42 | 2020-08-13 21:00 | 2.5 |
| WRF-P2 | one-way | Perimeter-2 | 2020-08-14 15:54 | 2020-08-14 15:00 | 2.5 |
| CFB-P1 | one-way | Perimeter-1 | 2020-08-13 23:42 | 2020-08-13 21:00 | 2.5 |
| CFB-P2 | one-way | Perimeter-2 | 2020-08-14 15:54 | 2020-08-14 15:00 | 2.5 |

of the models and thus a good first step to assess the consistency of the simulations. In the following experiments, UFS-2way and WRF-2way, the feedback from the fire to the atmosphere is activated. No simulation is performed with the standalone CFBM since the standalone code can only run in one-way mode.

In the remaining experiments, the fire is ignited from an observed perimeter. The UFS-P1, WRF-P1, and CFB-P1 are initialized from Perimeter-1, whereas UFS-P2, WRF-P2, and CFB-P2 are initialized from Perimeter-2. These six experiments do not pass information from the fire to the atmosphere (one-way coupling) since the standalone code only runs in one-way mode and the objective is to compare the consistency of the simulations. The perimeters were ignited at their corresponding timestamps, and the atmospheric state was initialized according to the closest model cycle used for initial condition (3-hourly intervals),

i.e., 2020-08-13 2100 UTC for Perimeter-1 and 2020-08-14 1500 UTC for Perimeter-2.

    The atmospheric models WRF and UFS ran with initial and boundary conditions from the High-Resolution Rapid Refresh (HRRR, Benjamin et al., 2016; Dowell et al., 2022) model at 3-hourly intervals. The simulations were configured with a single domain covering the state of Colorado at 3 km horizontal grid spacing. For WRF, we used a fire grid refinement of 30 to reach a 100 m grid spacing in the fire grid. A smaller domain centered over the fire with 100 m grid spacing was used when the CFBM

was involved (UFS and CFB experiments in Table 3). The atmospheric data necessary to run the CFB experiments came from the equivalent one-way WRF experiment. In all the simulations, we used the Anderson 13-fuels obtained from the LANDFIRE database.

    Although we configured the models as similarly as possible, some differences in the model configuration remained, including:





– The WRF model was configured with a fixed 12-s time step and 45 vertical levels. The physics processes were modeled with the following parameterizations: Thompson microphysics (Thompson et al., 2008); the Rapid Radiative Transfer Model for Global models (RRTMG, Iacono et al., 2008) and Dudhia scheme (Dudhia, 1989) for long and shortwave radiation, respectively; Mellor-Yamada Nakanishi and Niino Level 2.5 (Olson et al., 2019) for the planetary boundary layer parameterization; the revised MM5 surface layer scheme (Jiménez et al., 2012), and the Noah Land Surface Model

(Tewari et al., 2004).

– The experiments using the standalone CFBM (CFB-1way, CFB-P1, and CFB-P2) use atmospheric data from the WRF model. The atmospheric fields were provided in 20-min intervals. Effectively, this means the atmosphere runs with an identical configuration used by the WRF simulation, whereas the fire behavior component receives updates at 20-min intervals.

– The UFS model was configured through the Short-Range Weather App. The model time step and vertical levels were automatically set through the SRW workflow to 36 s and 65 levels, respectively. The atmospheric grid was set to a domain covering Colorado at 3 km grid spacing created specifically for this test case and available through the SRW App. We used 3 km grid spacing for the comparison because this is a standard grid spacing used at the National Oceanic and Atmospheric Administration (NOAA) for high resolution simulations with UFS (e.g., the Rapid Refresh Forecasting

System). It will be shown that at 3 km grid spacing the fire feedback to the atmosphere produce a small impact. However, the impacts are sufficient to assess the consistency between WRF and UFS fire simulations which is the main objective of this experiment. The physics parameterizations were set to the CCPP FV3-HRRR.

Finally, we set the fire wind height to 2.5 m in the WRF and CFBM simulations, whereas in UFS we set it to 5 m. This is motivated by our finding that the UFS wind speeds were weaker than WRF's for this case (see Figure 5 and related discussion

below); and since wind is one of the primary parameters controlling the rate of spread, choosing a height of equivalent wind speeds enabled us to compare the fire behavior evolution in a similar experiment. This enabled us to better assess the consistency of the fire behavior, by reducing WRF and UFS's differences originating in the atmosphere.

## 4.2 Results

### 4.2.1 Point ignition: one-way simulations

Figure 4 shows a comparison of the simulated fire area for the one-way simulations that start the fire from a point ignition (UFS-1way, WRF-1way, and CFB-1way) and the observed perimeters from CO-WIMS. As expected, the three simulations, WRF-1way (black), UFS-1way (blue), and CFB-1way (offline coupling of WRF and the fire behavior model, green) show consistency in the simulated fire perimeters. A perfect match is not expected considering WRF and UFS are different atmospheric models and the standalone code only uses a fraction of the WRF atmospheric data and even a different interpolation

from the atmospheric grid to the fire grid. In spite of these differences, the perimeters show good consistency, suggesting a proper implementation of the fire behavior model in the CFBM and its coupling with UFS. The simulated perimeters tend to



underestimate the fire rate of spread. The underestimation is likely affected by inaccuracies in the ignition time and location
records, and on the perimeter timestamp used to verify the simulations.

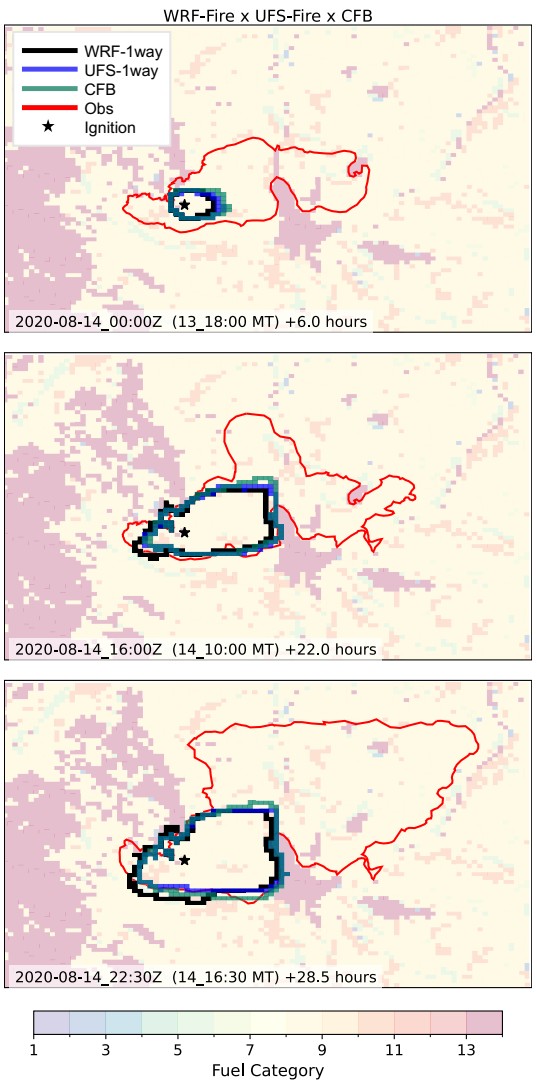

**Figure 4.** Simulated fire perimeters from experiments WRF-1way (black line), UFS-1way (blue line), and CFB-1way (green line) at the three
times with available observations. The observed perimeters are also shown (red lines).

The consistency of the simulations is further illustrated in Figure 5 which shows the evolution of the mean wind at the
fire perimeter and the burned area. The mean wind at the fire perimeter (Fig. 5a) is consistent in the three simulations (UFS-
1way, WRF-1way, and CFB-1way). The mean wind speed is mostly between 2 and $4\,\mathrm{m\,s}^{-1}$ and the simulations show similar
variability. Again, discrepancies between the CFB-1way and WRF-1way experiments are expected because we use different
interpolations methods to interpolate the wind speed to the target location, and the WRF data is updated only every 20 min





in the CFB-1way experiment. Discrepancies between UFS-1way and WRF-1way experiments are also affected by different
interpolation methods, but, more importantly, by different dynamical cores and parameterizations as was already pointed out.
However, in spite of these differences, we expected similar wind evolution in both simulations because we used the same
HRRR forcing for initial and lateral boundary conditions. This agreement is evident in Figure 5 (right). The wind consistency
translates into a similar evolution of the burnt area (Fig. 5, left). The similarities are remarkable considering the heterogeneity
of topography and fuels in this fire.

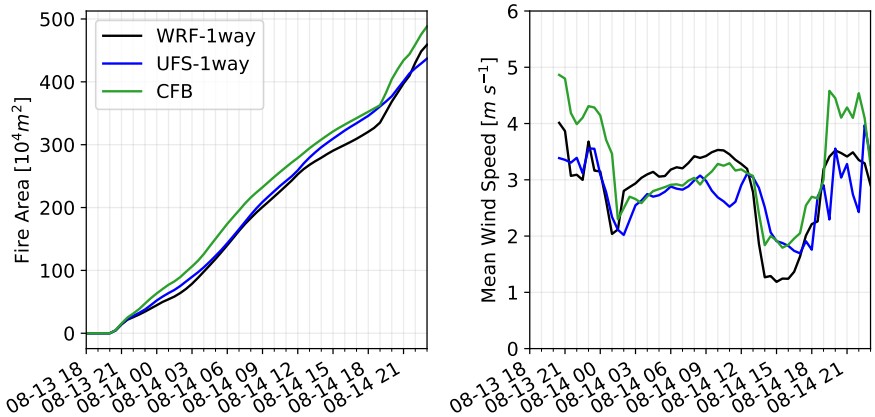

**Figure 5.** Evolution of the burnt area (left) and mean wind speed at the fire front (right) for experiments WRF-1way (black line), UFS-1way
(blue line), and CFB-1way (green line). The wind speed corresponds to the wind at 2.5 m above ground level for WRF and the standalone
model, and at 5.0 m above ground level for the UFS simulation.

As has been already mentioned, in order to obtain consistency in the simulated winds from WRF and UFS we had to use a
different height for the winds driven the fire in the models. This is illustrated in Figure 6, which shows the averaged wind at
the fire front from WRF-1way, UFS-1way and an equivalent simulation to UFS-1way except with the height set to 2.5 m as in
WRF-1way. The UFS winds are clearly biased low in comparison with WRF when both models use 2.5 m. However, changing
the height to 5 m in UFS allowed for having similar winds in both models which ultimately led to the agreement found in the
fire simulations (Fig. 4 and Fig. 5a).



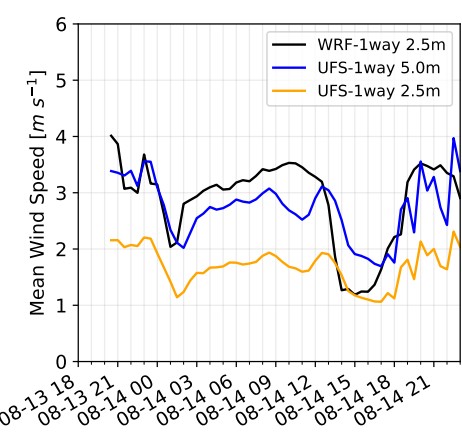

**Figure 6.** Averaged wind speed at the fire front from WRF at 2.5 m above ground level, and UFS at 2.5 m and 5 m above ground level.





### 4.2.2 Point ignition: two-way simulations

After ensuring consistency of the one-way coupled simulations, we turn our attention to the two-way coupled experiments (WRF-2way and UFS-2way). The comparison of the simulated and observed fire areas is shown in Figure 7. Again, we see an underestimation of the observed burned area with consistency between the simulated perimeters using WRF and UFS. Indeed,

results are similar to the ones obtained with the one-way experiments (Fig. 4). This is the first evidence of a relative small impact of the heat and moisture fluxes from the fire for these experiments that use 3 km grid spacing. In this direction, we further show the evolution of the burnt area and the wind speeds calculated with the 1way and 2way experiments using WRF and UFS (Fig. 8). The two-way experiments show small differences with respect to their one-way counterparts. Again, there is consistency between the two-way results for both the evolution of the wind speed at the fire front and the burnt area.

The consistency is also evident in the time series of the fire heat and moisture fluxes, and the smoke emissions (Fig. 9). These are the variables that are passed from the fire to the atmosphere component. This figure confirms that the simulations in one way and in two way produce similar fire fluxes at this grid spacing. There are more differences between the fluxes from WRF and UFS, but the fluxes from both models show similar variability which is what would be expected given the similar configuration of the atmospheric models and the methods used in the fire behavior models. The large fluctuations observed

in the UFS-2way experiment are due to the differences in time steps used by the CFBM and WRF-Fire. The CFBM operates with a time step of 0.5 s, whereas WRF-Fire uses an 18-s time step. This smaller time step causes the UFS-2way experiment to output instantaneous fluxes every 0.5 s, instead of averaged fluxes over the atmospheric model time step (which are used internally and not exposed to the fire output), as WRF-Fire does. Consequently, the UFS-2way experiment shows a large amount of fuel burned in a short period. This phenomenon is reduced when the fluxes were averaged over a longer period,

similar to the WRF-Fire timestep (not shown).

The consistency of the fluxes passed to the atmospheric component are illustrated with the differences in the heat fluxes calculated with the two way and one way experiments using WRF and UFS shown in Figure 10. The WRF differences are positive because WRF has a dedicated variable for the fire fluxes in the atmospheric grid and the fact that the fluxes are zero in the one way experiment. This is not the case for UFS because the fire fluxes are incorporated into the heat and moisture fluxes

at the surface. As a consequence, UFS differences sometimes can be also negative. However, at the location of the fire both models show positive differences of similar magnitude at the three times shown.

To conclude our analysis of the two way experiments we show in Figures 11 and 12 the impacts of the fire fluxes in the first model vertical layer temperature and moisture, respectively. Outside of the active fire area, there is a mix of positive and negative differences without a clear pattern. However, the temperature shows positive values over the active fire region,

increasing with lead time. The water vapor at the first model layer shows a smaller response to the fire moisture flux and no noticeable differences between the two way and one way simulation are seen as revealed by the near zero differences. Both models, WRF and UFS, show consistency in the spatial patterns of the differences.





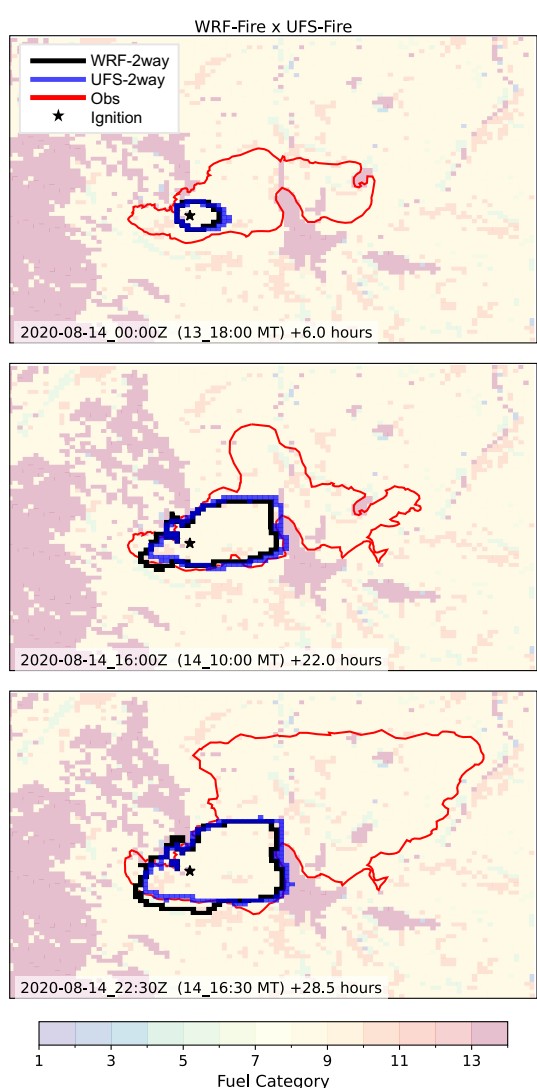

**Figure 7.** Same as Figure 4 but for the WRF-2way and UFS-2way experiments. Results for the CFBM are not shown because the model can not be run with two-way coupling.



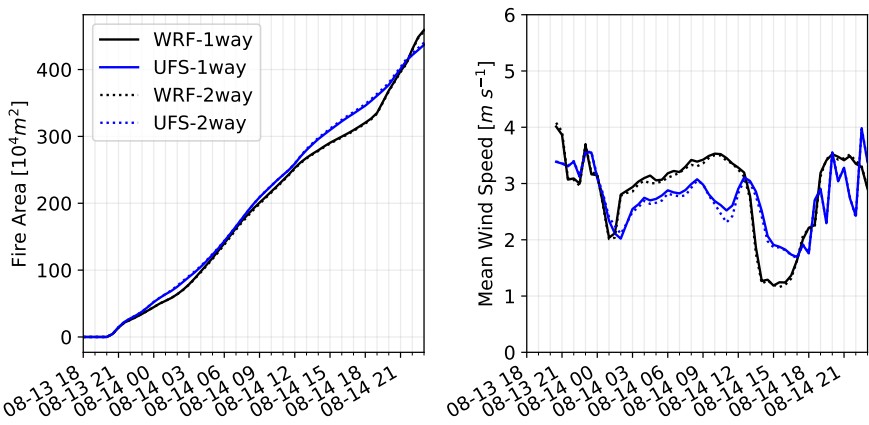

**Figure 8.** Same as Figure 4 but including both one-way and two-way coupled fire-atmosphere simulations with WRF and UFS.

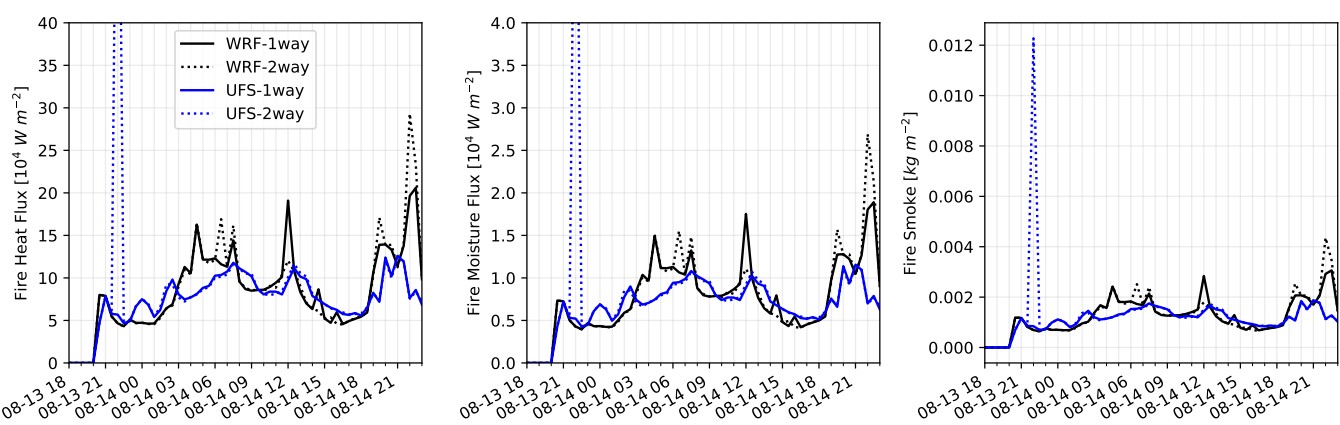

**Figure 9.** Time series of the fire heat flux (left), moisture flux (center), and smoke emissions (right).



**Figure 10.** Fire heat flux differences between two-way and one-way simulations using the WRF model (left column) and the UFS model (right column) at 3 h (top row), 6 h (middle row), and 12 h (bottom row) into the simulations.





**Figure 11.** Same as Figure 10 but for the temperature at the first model layer.



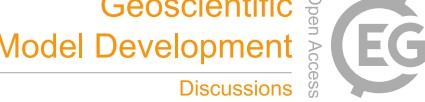

**Figure 12.** Same as Figure 10 but for the water vapor mixing ratio at the first model layer.





### 4.2.3 Ignitions from observed perimeters

The comparison of the simulated fire perimeters for those experiments starting from Perimeter-1 or Perimeter-2 against the
CO-WIMS observations is shown in Figure 13. For the experiments starting from Perimeter-1 (WRF-P1, UFS-P1, and CFB-P1) the fire mostly grows toward the northeast (Fig. 13 top). This spread is simulated accurately in the northern and the eastern parts of the fire, but results in a clear overestimation towards the northeastern portion of the perimeter. In the opposite side of the perimeter, in the southwest, there is an overestimation of the perimeter which seems to be a result of the fire perimeter not being active at the time of ignition as revealed by the similar observed perimeters in this region at the beginning of the
simulation and at the time shown (gray and red lines). If this is not the case, the discrepancies likely arise as misrepresentations of the winds in this portion of the perimeter. The perimeters from WRF and UFS are again consistent with a slightly faster spread of UFS in the northern portion. This is also the case of CFBM although in this case there are larger differences in the northwestern portion of the perimeter likely a result of the frequency of the atmospheric updates in CFB-P1.

The simulations starting from Perimeter-2 are also consistent with each other (Fig. 13 bottom). The three simulations (WRF-
P2, UFS-P2, and CFB-P2) show the fire expanding in the same directions. There is good agreement with observations in the western half of the perimeter. This time there is an overestimation in the southeastern part of the perimeter which is a result of misrepresentations of fuel barriers in the area, e.g. Highway 14 and Wright Creek, which run in parallel with the southeast portion of Perimeter-3 (red line) in this part of the fire front. To the northeast, the rate of spread of the fire is underestimated which is the opposite of what we found in this portion of the perimeter in the simulations starting from Perimeter-1 (Fig. 13
top). This could be attributed to timing errors in the simulation of the wind field, or inaccuracies in the timestamps of the observed perimeters. In any case, there is no evidence of a systematic bias in the rate of spread for this case. The CFBM simulation appear to spread faster in some portions of the perimeter highlighting the effects of the frequency at which WRF data is updated.



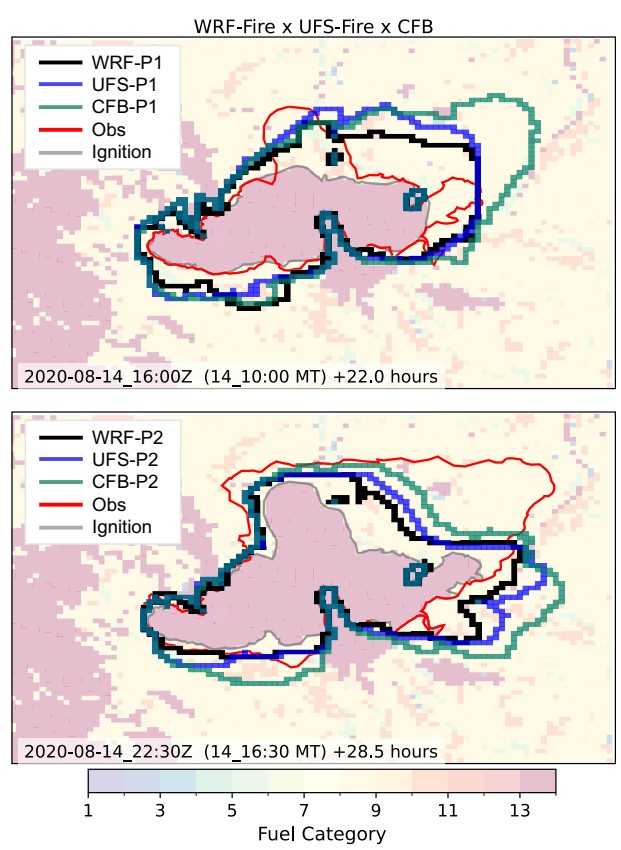

**Figure 13.** Simulated perimeters for the WRF-P1, UFS-P1, and CFB-P1 experiments at the time of the observed Perimeter-2 (top) as well as the simulated perimeters for the WRF-P2, UFS-P2, and CFB-P2 experiments at the time of the observed Perimeter-3 (bottom). The observed perimeters at the beginning of the simulations (gray line) and at the valid time of the simulations are also shown (red line).



## 5 Conclusions

In this study we present the CFBM, which at its core in its current version 0.2.0 is a redesigned implementation of WRF-Fire 4.3.3 procedures in a new model available for coupling with atmospheric models through the ESMF library. The use of the ESMF library minimizes the code needed to interpolate variables between the atmospheric and fire grids, and the synchronization and exchange of data between the atmosphere and the fire components. The fire behavior model is available as a NUOPC and thus follows ESMF standards to integrate Earth system components. Coupling to already existing Earth system
models with the atmospheric component available as a NUOPC should be straightforward. In its simplest configuration, the atmospheric model passes the winds and the roughness length to propagate the fires (one-way coupling). If the evolution of the ground FMC is simulated, other standard surface variables (pressure, temperature, humidity, and precipitation) are used by the fire model. In addition, the fire behavior model can provide feedback to the atmosphere via sensible heat and latent heat fluxes, and smoke emissions (two-way coupling). This configuration requires connecting the heat/moisture fluxes to the atmospheric
tendencies in the atmospheric model to modify the atmospheric evolution which in turn affects the winds and the fire evolution accounting for fire-atmosphere interactions. Smoke emissions simulated as a fraction of the fuel burned at the ground are also available to the atmospheric component for coupling with radiation and cloud processes if desired in the future.

In order to facilitate the evaluation of the CFBM NUOPC, we closely followed WRF-Fire version 4.3.3 methods in the CFBM version 0.2.0. This allows for comparing CFBM simulations with an atmospheric host to WRF-Fire, a state-of-the-art
fire behavior model. In spite of following WRF methods, substantial changes have been introduced in the code to create a fire model independent of the atmospheric component that allows for ESMF coupling, improved performance, and readability, as well as facilitating maintenance and extension of the code. The model can be also run in standalone mode using data from an existing WRF simulation. This standalone version of the code does not require the ESMF library and can be used to test developments and sensitivities to the fire evolution.

The CFBM has been coupled to UFS. This allowed us to compare UFS simulations to WRF-Fire to ensure a proper development. Our results for the Cameron Peak Fire show consistency of the fire evolution simulated by the standalone model, UFS, and WRF-Fire. Comparisons of one-way versus two-way fire simulations showed small impacts in the atmospheric evolution. This is mostly a consequence of the relatively coarse grid spacing used, 3 km, but it was sufficient to ensure consistency between UFS and WRF-Fire. The consistency between UFS and WRF-Fire is encouraging for the UFS community to start per-
forming fundamental research on fire-weather interactions. This consistency is also the starting point of ongoing developments to improve the accuracy of the simulated fire evolution beyond the original WRF-Fire model.

The CFBM is the first fire behavior model available for coupling with atmospheric models as a NUOPC in ESMF. This is expected to facilitate its adoption by other atmospheric models, especially those already using the ESMF library. In this way, the fire community can benefit from having the same modeling framework. We also envision that our efforts may pave the
way for future fire model intercomparison efforts, a common practice in various atmospheric science modeling communities. Indeed, our vision is to foster collaborative development in fire behavior modeling with the ultimate goal of increasing our fundamental understanding of fire science and minimizing the adverse impacts of wildland fires.



*Code availability.* The CFBM is available as open source code in Zenodo (https://zenodo.org/records/13357368) and in this git repository https://github.com/NCAR/fire_behavior. WRF is available at https://github.com/wrf-model and the code for the UFS model is available from this git repository: https://github.com/ufs-community.

*Author contributions.* PAJM, ME, DR, MF, and TWJ created the CFBM model and its NUOPC. ME, DR and PAJM coupled it to UFS. MK added the CFBM to the UFS SWR App. PAJM and MF conceptualized the experiments. MF run all the simulations and created the figures. PAJM prepared the article with contributions from all the co-authors.

*Competing interests.* The contact author has declared that none of the authors has any competing interests.

*Acknowledgements.* We would like to thank Dr. Sudeer Bhimireddy, Dr. Branko Kosovic, and Dr. Ravan Ahmadov for feedback provided during the development of the CFBM. This material is based upon work supported by the NSF National Center for Atmospheric Research, which is a major facility sponsored by the U.S. National Science Foundation under Cooperative Agreement No. 1852977. Funding for ME, MF, MK, PAJM, and TWJ was provided by NOAA under Award No. NA22OAR4590514-T1-01. The simulations were conducted on Derecho (https://doi.org/10.5065/qx9a-pg09), which is provided by NSF NCAR's Computational and Information Systems Laboratory (CISL), and sponsored by the U.S. National Science Foundation.





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
