# Peer review of "The Community Fire Behavior Model for coupled fire-atmosphere modeling: Implementation in the Unified Forecast System"

_Geoscientific Model Development, 2024_

## Referee Comment (RC3)

**Review for "The Community Fire Behavior Model for coupled fire-atmosphere modeling: Implementation in the Unified Forecast System"**

The manuscript introduces the Community Fire Behavior Model (CFBM), a newly developed fire behavior model designed for seamless coupling with different atmospheric models using the Earth System Modeling Framework (ESMF). The key objective of CFBM is to provide a flexible and modular framework for simulating coupled fire-atmosphere interactions without requiring model-specific interpolation procedures. This approach is intended to foster broader collaborations and model integrations beyond the traditional Weather Research and Forecasting (WRF) model with fire extensions (WRF-Fire).

The current review starts with specific comments in each section of the manuscript and finishes with a general comments section, followed by a final decision section.

The fire behavior model:
- The input data interface remains dependent on Geogrid from WPS, which necessitates pre-processing via WRF. Consider addressing whether the model can be decoupled from WRF-specific tools.
- The results could benefit from improvements to the fire initialization since the method presented seems to mismatch fire and atmosphere dynamics. The authors should consider using a spin-up period during the perimeter initialization.
- The two available options for fire wind interpolation require further explanation. Additionally, if wind reduction factors are applied, their implementation and rationale should be discussed.
- The manuscript should specify whether Lambert-Conformal projection is the only supported coordinate system in CFBM. If additional projections are supported, this should be clarified.

Experimental setup:
- The downscaling from a 3 km atmospheric grid to a 100 m fire mesh presents notable limitations:
    - The atmospheric resolution is too coarse to adequately capture fire-atmosphere interactions.
    - The fire mesh resolution is also too coarse to properly resolve fire spread parameterizations.
- The rationale behind different configurations between WRF and UFS regarding vertical layers and time steps should be justified. For instance, the use of different number of vertical layers and time steps?

Validation:
- Adjusting wind height to better match fire progression raises concerns about the validity of the model's predictive capability. It should be clarified whether this adjustment is an empirical correction or an inherent part of the modeling approach.

- The similarity of results between one-way and two-way coupling suggests that the two-way coupling mechanism has not been adequately validated. Further evidence of feedback interactions is needed.
- The validation would benefit from quantitative evaluation metrics such as the Jaccard index or Sørensen–Dice coefficient to assess model performance systematically.
- While the manuscript acknowledges underestimating fire spread in some regions and overestimating in others, it does not provide sufficient explanation for these discrepancies. A sensitivity analysis on parameter uncertainties (e.g., fuel properties, and wind corrections) would strengthen the discussion.

Other Minor Comments:
- Line 156: The phrase "substantial portions of new code" requires further clarification regarding the specific novel contributions.
- Line 166: The term "atmospheric dependencies" is ambiguous and requires clarification.
- Lines 166-167: The phrase "the WRF grid to the WRF-Fire grid" is misleading, as both the atmospheric and fire grids are components of WRF-Fire.
- Lines 167-168: The statement "The calculation of the fire grid latitude and longitudes..." appears misplaced within the section.
- Lines 168-169: The manuscript does not adequately describe the improved approach for geolocation determination or how it enhances accuracy.
- Line 174: The methodology for handling grid mismatches and specifying the fire domain should be explicitly detailed.
- Line 176: ... the three-dimensional horizontal wind components (U and V variables in WRF).

General Comments:
- The manuscript presents limited innovations in fire physics, appearing to largely replicate WRF-Fire, with minor modifications following Mandel et al. (2011) and Muñoz-Esparza et al. (2018). The novelty of the contribution should be better articulated.
- The study is restricted to a single fire event (Cameron Peak Fire), which limits the generalizability of the findings. Expanding the analysis to multiple fire events with varying topographies and meteorological conditions would improve the robustness of the conclusions. If additional events cannot be incorporated, at a minimum, a sensitivity analysis should be conducted.
- The computational trade-offs of coupling CFBM with other atmospheric models should be acknowledged and compared to existing solutions such as WRF-Fire.

Final Decision:
The manuscript is not recommended for publication mainly for the following reasons:

1.  Misalignment with the journal's scope: The manuscript presents a well-structured refactoring of the WRF-Fire codebase to create a more modular and flexible fire behavior model. While the core fire physics remains largely unchanged, the restructuring enhances model interoperability, making it easier to couple with different atmospheric models using ESMF. This modularity is a valuable contribution to fostering broader collaborations within the fire modeling community. However, for this journal, which emphasizes advancements in geophysical modeling, the paper would benefit from further developments that go beyond code refactoring, such as improvements in fire physics or additional parameterizations. The reviewer suggests submitting this manuscript to another journal with a different scope after resolving the issues in the following point.
2.  Issues with validation methodology:
    a.  Discrepancies in model configurations (e.g., different vertical levels and parameterizations).
    b.  Inappropriate scales that likely weaken fire-atmosphere feedback.
    c.  Validation primarily relies on qualitative comparisons, which are insufficient for rigorous model evaluation.

A final recommendation is that instead of using a real-world fire case with significantly different configurations and scales between the coupled components, the authors should consider an idealized test case that allows for a more controlled and rigorous evaluation of the coupling framework.

---

## Author Comment (AC1)

RESPONSE TO REVIEWERS COMMENTS

"The Community Fire Behavior Model for coupled fire-atmosphere modeling: Implementation in the Unified Forecast System" by Pedro A. Jiménez y Muñoz, Maria Frediani, Masih Eghdami, Daniel Rosen, Michael Kavulich, and Timothy W. Juliano.

—------------------------------------------------------------------------

Reviewer #1

*Review of "The Community Fire Behavior Model for coupled fire-atmosphere modeling: Implementation in the Unified Forecast System" by Pedro A. Jiménez y Munñoz, Maria Frediani, Masih Eghdami, Daniel Rosen, Michael Kavulich, and Timothy W. Juliano.*

*The manuscript presents a fire behavior model that can be coupled with the existing atmospheric models. The study presents results when the community fire behavior model is coupled with the Unified Forecast System for fire spread episode focused during Summer of 2020 in Colorado. The model is run in a standalone model, coupled mode and the results are compared against the WRF-Fire model predictions.*

*Overall, the manuscript is well written and the fire behavior model is well explained. Figures and model schematic depictions are clear. The study is valuable as the CFBM model presented could be run with a user-selected atmospheric model as long as the required variables are present. The manuscript is suitable for publication after revisions addressing the below comments.*

RESPONSE

We would like to thank this reviewer for the time she/he devoted to revise the manuscript and for the positive perspective about the manuscript. We have improved our explanations and revisited the experimental set up in order for UFS and WRF to have the same set of parameterizations, time step, and number of vertical layers. We have also increased the grid spacing from 3 km to 1 km. As a result, there is a better agreement with the winds from UFS and WRF and the simulated perimeters. We have also introduced quantifications of the agreement between the simulated perimeters. Below, we have reproduced the comments of this reviewer and we have provided a detailed answer indicating how we have modified the original version of the manuscript. We believe the modified version of the manuscript is a stronger contribution.

COMMENT 1

*Comments:*

*1.      Expand or describe the wind and slope correction terms mentioned in Equation 1. Without the wind correction term formula, I am having difficulty understanding why the user would have to select a fire wind height, when it is much simpler to use the 10-m winds (which are usually available from many atmospheric model outputs) as the driving force for the fire perimeter. Also, is there an upper limit for the rate of spread in the model to address any unrealistic values or sudden spikes such as the one observed for fire heat/vapor fluxes and emissions shown in Figure 9?*

RESPONSE

These terms involve complicated parameterizations and adding them would be probably too much level of detail and thus we prefer to cite the relevant document with all the detailed explanations: "*A comprehensive description of each term involved in the parameterization is provided by Andrews (2018)*". The height selected by the user represents the mid-flame height. We have added this information to the new version of the manuscript. Imposing the height of the winds that drive the fire is the only option in WRF-Fire and that is why we used it given the main objective of testing the consistency of our implementation in UFS-CFBM. Other options involve the use of wind adjustment factors (WAF), or in other words, using a different height for each of the fuel types depending on the fuel characteristics. The use of WAFs in combination with the 10 m winds, as the reviewer suggested, is an attractive option for extensions of the CFBM and we are currently testing this option (Eghdami et al. 2025). We have clarified these aspects too: "*In the future, we would like to alleviate this subjective choice and implement the use of wind adjustment factors to automatically identify the height that drives the fire evolution based on the fuel characteristics (Eghdami et al. 2025).*". The upper limit of the rate of spread is 6 m/s and we now say this in the new version of the manuscript: "*The fire rate of spread is limited to a maximum of 6 m/s*". The spikes in the old Figure 9 were related to a different aspect of our previous experimental set up (the time step) that we have fixed in our new experimental set up (new Figure 10 in the manuscript).

COMMENT 2

*2.        Since the WRF model and the UFS model were initialized from HRRR, why is there such a bias in the wind speed predicted by UFS? Did the authors perform any sensitivity analysis to identify the source of this discrepancy? Can we expect UFS to underestimate the wind speed in general? Are there other studies in the literature that pointed this out? Choosing a different height simply to match the WRF wind speed may not fully address the complexities involved, especially when the height is user given. As the authors mentioned, the choice of fire wind height is one of the key input parameters in running the model. Further analysis could help the CFBM users to understand the uncertainties involved and make an educated choice of the fire wind height. If the intention is to simply use the WRF-Fire model as the ground truth or reference, use of different physics parameterizations in UFS compared to WRF would obviously yield different results. Also, as mentioned near Line 105; how will the model perform if (say) 10m winds are interpolated to 2.5m or 5m based on the logarithmic profile.*

RESPONSE

In the new version of the manuscript we have resolved this discrepancy. It came for the use of different land surface models which affects the roughness length which strongly conditions the magnitude of near surface winds. The wind speed from WRF and UFS are in much better agreement now (new Figure 7). The correlation, bias, and mean absolute error between the wind speed from UFS and WRF is 0.69, -0.3 m/s, and 0.5 m/s, respectively. We have added this information to the manuscript. In the future we would like to avoid the user having to select the height using WAFs (see answer to previous comment). Using the 10 m winds in combination with the logarithmic profile is not going to change results much and that is why we would like to use the 10 m winds in the future. For now we use the option available in

WRF-Fire which allows for a clean comparison of the models to ensure the adequacy of our implementation.

The other option in WRF-Fire, which we did not test here but we describe, is to use the winds from higher vertical levels and the logarithmic interpolation to decouple the impacts of the fire in the atmosphere since the Rothermel parameterization of the rate of spread was developed for using ambient winds to drive the fire evolution.

COMMENT 3

*3.        It is very surprising that the model heat flux value differs outside the fire perimeter, especially in the UFS runs (Figure 10). Even more surprising is the presence of negative values in the UFS heat flux differences. If the only difference among the coupled and uncoupled simulations is the feedback from the fire pixels to the atmosphere, why will there be any changes to the surface heat flux far away from the fire perimeter! For the purpose of comparison, in Figure 10, it would be better if the WRF variable also includes the total surface heat flux, as the uncoupled values are subtracted from the coupled, only the fire induced heat flux would remain.*

RESPONSE

Only UFS shows this behavior because there is only one heat flux that includes the standard head flux from the land surface model plus the fire contribution. The positive/negative values outside the fire perimeter result from subtracting the fluxes from the one-way simulation to the two-way simulation and nonlinear effects in the atmosphere introduced by the fire which affects the standard land surface model heat flux. This is not the case for WRF-Fire (the values are always positive) that has a dedicated variable for the fire heat flux (that is zero for an uncoupled simulation). Hence for WRF-Fire, only the fire induced heat flux is shown. This is the cleanest comparison we can provide. We have clarified this in the version of the manuscript.

COMMENT 4

*4.        Interesting to see the mean wind speed unaffected even if the temperature increased in both WRF and UFS models as a result of 2-way coupling, which would change the vertical velocity in the model. It would be better to show the change in vertical velocity due to 2-way coupling and it could be used to justify the insignificant effect of 2-way coupling on the fire spread and mean winds shown.*

RESPONSE

We have calculated the impacts in the vertical velocity near the ground. See Figure 1 of this document. There are no noticeable impacts in the first few hours of the simulation, but when the fire grows, 12 h into the simulation, we see, as expected, an increase in the vertical velocity in both WRF and UFS. We have included this figure in the new version of the manuscript.

**Figure 1**: Fire heat flux differences between two-way and one-way simulations using the WRF model (left column) and the UFS model (right column) at 3 h (top row), 6 h (middle row), and 12 h (bottom row) into the simulations. To illustrate the location of the fire we also show Perimeter 3 valid for August 14 2216 UTC (red solid line).

COMMENT 5

*5.      It is hard to follow the discussion near Line 385, about the large fluctuations in fluxes and emissions in the UFS-2way run. Why would the WRF-Fire time step 18s be relevant to the fluctuations in UFS output? Also, I thought WRF used a fixed 12s timestep! Why do the large fluctuations in UFS-2way variables start 3 hours after the fire initialization?*

RESPONSE

We have rerun the simulations in the new version of the manuscript using 1 km grid spacing and a time step of 4 s in all the simulations. The unclear explanations are no longer needed since now there is a much better agreement between the simulations. There are still peaks of smaller magnitude and these peaks do not need to perfectly align because UFS and WRF use different dynamical cores and the atmosphere is nonlinear which affects the winds and the fire evolution. However, we would expect to

see good resemblances in the lower frequency evolution of the fire fluxes from both models, and that is what we see in the new version of the figure (correlation 0.77) with the time series of the fluxes (see new Fig. 10 and updated discussion).

COMMENT 6

*6. Line 421: It's hard to justify this line: "In any case, there is no evidence of a systematic bias in the rate of spread for this case." From Figures 5 and 8, the fire area from UFS (1-way or 2-way) seems to be larger than the WRF simulated area. This is counterintuitive when one takes into account the mean wind speed differences. For a good portion of the simulated time, the mean wind in UFS is weaker than in the WRF and yet the fire area in UFS is larger than in WRF. Authors could add the timeseries of the difference between the fire area simulated by UFS and WRF for the -P1 and -P2 cases to justify if there is a systematic bias.*

RESPONSE

This sentence referred to the results for the simulations starting from fire perimeters. We have rerun our simulations to have a better agreement between WRF and UFS configurations. We now have the same set of parameterizations, time step, and number of vertical layers. There is much better agreement now in the simulated perimeters for UFS and WRF for the -P1 an -P2 cases. We have updated our explanations and removed that sentence in Section 4.2.4.

Also in the new Figures 5 and 8, now Figures 7 and 9, there is good agreement between the UFS and WRF winds. The correlation, bias, and mean absolute error between the wind speed from UFS and WRF is 0.69, -0.3 m/s, and 0.5 m/s, respectively. Note that we are showing the wind averaged over the complete fire perimeter and there will be regions where the wind points towards the burned area and this wind speed does not contribute to advancing the fire perimeter. See the updated description of the figures.

COMMENT 7

*7. For the 2way runs, please add details about which layer the smoke emissions area added and that the smoke is being added as a passive tracer, i.e., it does not carry any thermodynamic or chemical properties. Even better would be to show a 3D visualization of the fire progression with smoke tracer and stream lines showing any updrafts over the fire.*

RESPONSE

We have clarified our explanations of the smoke tracer: "*The smoke emissions are added to the first atmospheric layer of a passive tracer to represent the smoke transport and dispersion associated with atmospheric dynamics.*"

COMMENT 8

*8. Discuss any limitations of the CFBM. For example, can it be coupled with very high resolution models (grid size less than 100m or large-eddy scales)?*

RESPONSE

Yes, the fire model can be run with the atmosphere configured in large-eddy simulation (LES) mode as it has been done before with WRF-Fire (e.g., Jimenez et al. 2018). Currently, the main limitation of CFBM with respect to other fire models is that the model can run in only one cpu. We are working on adding the parallelization. We already mentioned this in the manuscript but we now say it one more time: "*Currently, the main limitation is that CFBM can run with just one dynamical core but we are working to include OpenMP parallelization soon.*"

COMMENT 9

*9. Line 294: Expand on or use a reference for the line "And third, we updated the VEGPARM.TBL to correct for a bug in the table." What bug?*

RESPONSE

There were some missing or incorrect lines in the table. This is a fix from the official WRF release: https://github.com/wrf-model/WRF/commit/f0a4f0359aa28306ac2c59c559fed02db4ebf077

We now say "*And third, we updated the VEGPARM.TBL to correct for missing/incorrect lines, this fix being part of the official WRF release.*"

COMMENT 10

*10. Use consistent references, in the discussion around Figure 5, at places the subplots were referred to as Figure 5a and later in the text, they were referred to as Figure 5 (right/left).*

RESPONSE

Thanks, we have corrected this.

COMMENT 11

*11. Describe the red perimeter line in Figure 10 in the caption. It would be better if Figures 10, 11 and 12 are shown for the same times as in Figures 4 and 7.*

RESPONSE

We have added the description in the caption of Figure 10. We show results for the first hours of the fire evolution, 3 h, 6 h and 12 h into the simulation to minimize the impact of nonlinear effects of the atmospheric evolution. The second one, 6 h into the simulation, is the same time as the first panel in Figures 4 (new Fig. 5 in the new version of the manuscript) and 7 (new Fig. 8).

COMMENT 12

*12. As this is similar to and compared with WRF-Fire in the manuscript, it would be useful for the end user to know about any benefits CFBM would have over other existing models such as WRF-Fire in terms of computational requirements.*

RESPONSE

The main benefit that we can think at this moment in terms of computational requirements is that the fire grid does not need to match the atmospheric grid. The fire grid could be much smaller. For example, one can run simulations over the Contiguous U.S. with UFS and simulate a fire in Colorado over a small domain like the one we show in the manuscript. This will introduce substantial computational savings. Also, the atmospheric grid could be complicated (e.g. unstructured), but the NUOPC cap will automatically perform the interpolation of variables between the two grids. We have highlighted these aspects: "*However, the region covered by the fire simulation in CFBM does not need to match the atmospheric domain allowing for using smaller fire domains which reduces the computational requirements of the fire model. Also, it is possible to have any kind of atmospheric grid (e.g., unstructured) and the NUOPC cap will perform the interpolation of variables automatically.*"

**References**

Andrews, P. L., 2018: The Rothermel surface fire spread model and associated developments: A comprehensive explanation, General Technical Report RMRS-GTR-371, US Department of Agriculture, Forest Service, Rocky Mountain Research Station, Fort Collins, CO, U.S.A.

Eghdami, M., P. A. Jimenez y Munoz, and A. DeCastro, 2025: Sensitivity to the representation of wind for wildfire rate of spread: Case studies with the Community Fire Behavior model. Fire, 8, 135.

Jimenez, P.A., D. Munoz-Esparza and B. Kosovic, 2018: A high resolution coupled fire-atmosphere forecasting system to minimize the impacts of wildland fires: Applications to the Chimney Tops II wildland fire event. Atmosphere, 9050197.

---

## Author Comment (AC2)

**RESPONSE TO REVIEWERS COMMENTS**

"The Community Fire Behavior Model for coupled fire-atmosphere modeling: Implementation in the Unified Forecast System" by Pedro A. Jiménez y Muñoz, Maria Frediani, Masih Eghdami, Daniel Rosen, Michael Kavulich, and Timothy W. Juliano.

—--------------------------------------------------------------------

**Reviewer #2**

General Comments:

*The paper presents the development and implementation of the framework that allows coupling between the surface fire model and the Unified Forecasting System (UFS). The proposed framework/coupler is presented as designed to enable coupling with various atmospheric models using the Earth System Modeling Framework (ESMF). The fire model itself is generally a copy of the WRF-Fire code packaged so that it can be run in standalone mode or coupled with atmospheric models like the Unified Forecast System (UFS). It includes features such as a dynamic fuel moisture content model, fire propagation based on the level set method driven by terrain and wind conditions, simplified fuel consumption, and emission modules that mimic the existing capability of the fire code in WRF.*

*The paper describes the model's structure, its coupling with UFS, and the results of simulations of the Cameron Peak Fire for cases initialized from a point ignition and fire perimeter.*

RESPONSE

We would like to thank this reviewer for the time she/he devoted to revise the manuscript. We have reproduced below each of the comments raised by this reviewer followed by a detailed answer indicating how we have changed the manuscript to accommodate the comment. Our main purpose with this contribution is to 1) present the model, the first fire model designed for interoperability with atmospheric dynamical cores (via the ESMF libraries, a standard approach for coupling Earth system components), and 2) to show the adequacy of our implementation by comparing WRF-Fire and the UFS coupled to the Community Fire Behavior model (CFBM). This last aspect is possible because we purposely follow WRF-Fire methods in our current implementation. We would not call CFBM a copy of WRF-Fire, we see it as a new model and we have highlighted why in this revision. We do, on purpose, rely on WRF–Fire methods, in this current version, to evaluate the adequacy of our implementation by comparing WRF-Fire and UFS-CFBM results. The WRF-Fire methods are mostly confined to the physics directory in Figure 1. The rest of the directories have mostly new code which is necessary to create the CFBM and make it available as a NUOPC component for coupling with other Earth system components. Also, most of the WRF-Fire methods are implemented as parameterization that can be easily changed, and we aim to do so in future versions of the model.

**COMMENT 1**

*At a fundamental level, the paper does not appear to meet the publication standards of GMD. The fire model presented is essentially a refactored version of an existing code, which replicates its shortcomings and issues without introducing substantial novelty. Additionally, the description of the fire model's structure lacks clarity,*

*which would hinder community-based development. It is crucial to clearly separate the fundamental processes involved, such as the representation of the fire front and its propagation, fuel consumption, emissions, fuel moisture, and rate of spread. Unfortunately, these aspects are not adequately addressed in the paper, undermining its potential as a foundation for a community model.*

RESPONSE

The work in this paper provides novel functionality not seen in the previous WRF-Fire model: we introduce an agnostic coupling layer, based on ESMF, that allows for a variety of atmospheric coupling capabilities including and not limited to UFS coupling, which currently includes the FV3 dynamical core and CCPP physics package seen in this paper. We indicate this in the abstract "*we have created, for the first time, a fire behavior model that can be connected to other atmospheric models without the need of developing specific low-level procedures for the particular atmospheric model being used*". The CFBM is the first fire model designed for interoperability with atmospheric dynamical cores. We would now downplay the value of this achievement.

Section 2.1 provides the information mentioned by the reviewer. We start this section with a brief introduction to the main concepts of CFBM. Then we describe the fire ignition, how it is represented, and how it is propagated. During this description we describe how the fuel is represented and how the rate of spread is calculated. We also describe how the fuel is consumed and the feedbacks to the atmosphere which include the emission of smoke. Hence, we describe all the aspects mentioned by the reviewer. We have carefully reviewed the section again and we have clarified the more obscure parts. To facilitate isolating the description of the fire model, we have also moved the description of the offline coupling with the atmosphere to the new section 2.2 (see answer to next comment). We believe all the relevant information is there, and readers should be able to follow the methods used to represent the fire and its evolution, but if the reviewer still thinks we should clarify any particular aspect we will be happy to do so.

COMMENT 2

*The description of the modeling system is challenging to follow and often appears disorganized. A more structured explanation, beginning with a high-level overview and then detailing specific components, would greatly assist the reader in understanding the overall architecture of the proposed system, including the coupling strategies. Additionally, a fundamental diagram illustrating the relationships between key components such as UFS, ESMF, NUOPC, CFBM, and the data flow in UFS, CFB, UFS-1way, and UFS-2way would be beneficial. The current use of diagrams needs improvement, as they do not effectively clarify the presented developments.*

RESPONSE

We have restructured Section 2 to better describe the fire model (see answer to previous comment), and the coupling to the atmosphere. To this end Section 2.2 is now called "Coupling with the atmosphere" and includes subsections 2.2.1 and 2.2.2 that describe the offline and online couplings, respectively. We have also moved some part of the discussion to Section 2.3 dealing with "Compilation and testing". We have also clarified our explanations. The first 3 Figures of the manuscript show the structure of the code and the data flow between UFS and the fire model. We have enlarged our explanations of the NUOPC which is essentially a lightweight

file, the cap, to connect models. NUOPC is part of ESMF. We have clarified these aspects in the reformulated Section 2. We believe our description should be much easier to follow now.

**COMMENT 3**

*Furthermore, the file system structure is incomplete; for example, it does not specify the location of configuration files. The coupling diagram also fails to indicate the data flow for the experiments presented. It is unclear how the one-way coupled simulations are performed and which components are used in their execution.*

RESPONSE

The only configuration files needed to run a simulation are the Geogrid file and the namelist. There are examples of the namelists in the testing directory (e.g., tests/test7/namelist.fire). This information was missing in the previous version of the manuscript. We have added the following to clarify this aspect: "*Besides the Geogrid file, the only other file needed to run the model is the namelist (i.e., namelist.fire). Examples of the namelist are provided in the tests directory, inside the directories for each test.*"

We now provide a better description of the data flow in the coupling diagrams (see more detailed information below).

With respect to our unclear explanations regarding the one way coupled simulations, we have clarified this in our extended and rearranged Section 2. More specifically, Section 2.2.1 describes the offline model that reads the WRF output to drive the CFBM, and Section 2.2.3 describes our wrapper to create the WRF-Data NUOPC component and how we test the NUOPC implementation. Please, refer to the answer to the previous comment as well.

**COMMENT 4**

*The language in the document requires some revisions. There are inconsistencies in the tense used throughout the writing, and many sentences could be rephrased for clarity. Additionally, shifting the focus from "we developed..." to "the module... has been developed..." would improve readability.*

RESPONSE

We have revisited the language and grammar with especial emphasis from one of us that has English as the first language. We believe the language should be appropriate for publication standards.

**COMMENT 5**

*At the technical level, the paper faces several significant issues, including a flawed design in its numerical experiments, a lack of scientific rigor in validation and coupling strategies, and insufficient detail for replication. Critical elements, such as domain sizes and placement, are omitted. Additionally, the plots depicting fire simulations are missing scales and axis labels.*

RESPONSE

We have revisited our configuration of UFS and WRF to have a very similar set up (same set of parameterizations, number of vertical layers, and time step), and we have included quantifications in the

comparison of model results (see details below), improved most of the figures, and provided the sizes and locations of the domains to facilitate replication of results. We believe the experimental set up and the results, including quantifications now, are sufficient to illustrate the consistency of WRF-Fire and UFS-CFBM models which is our main purpose in this manuscript. Our main emphasis in model intercomparison was not clear enough in our previous version of the manuscript, and thus our explanations of this aspect have been improved in the new version of the manuscript.

COMMENT 6

*Although UFS and WRF are different models, the parameterizations used in UFS are essentially a subset of those available in WRF. Therefore, it is important to focus on creating a comparable setup. Key aspects of the simulation, such as the time step and the number of vertical levels, could have been adjusted in WRF to align more closely with UFS, which would help reduce inconsistencies between the model configurations.*

RESPONSE

We have revisited our experimental set up and we now have the same set of parameterizations, time step, and number of vertical levels. This has helped to produce results from both models, UFS and WRF, that are in better agreement with each other.

COMMENT 7

*More attention must be paid to the ignition strategy. The issue of integrating fire observations into coupled fire-atmosphere models has been studied, and there are existing solutions that enable both smooth ignition and selective fire activation see for example Kochansky et al. 2023 (https://doi.org/10.3389/ffgc.2023.1203578). The instantaneous ignition method used in this study is shown to negatively impact model stability and although widely used for uncoupled models is not optimal for coupled simulations. Another issue is the fuel moisture. It remains unclear how it was preconditioned and why the fuel moisture model implementation was not evaluated.*

RESPONSE

We use the same fire initialization procedures in UFS-CFBM and WRF-Fire to check for consistency between results of both fire behavior models. The procedure to initialize the fire from a perimeter is the only option available in WRF-Fire. This methodology has been successfully used in previous research and thus supported by publications (e.g., DeCastro et al. 2022; Turney et al. 2023). We understand there are other initialization methods that can be explored in future versions of the CFBM (e.g., Kochanski et al. 2023) where our intention will be to go beyond current WRF-Fire methods. We clarified these aspects in the new version of the manuscript when describing the fire initialization methods: "*This fire-perimeter initialization method is the only available in WRF-Fire and has been successfully used in previous research with the WRF-Fire model (DeCastro et al. 2022; Turney et al. 2023). More sophisticated methods (Kochanski et al. 2023) could be explored in future versions of the model that will go beyond WRF-Fire methods.*"

Note that the simulations we present in the paper starting from an observed perimeter are one-way simulations in order to compare with the simulation we performed with the standalone version of CFBM that reads WRF data offline. The fire and atmospheric dynamics do not play a big role here. This experiment aims to check for the consistency of UFS and WRF-Fire when starting from a given fire perimeter, and our results support the consistency between the models.

We set the FMC constant to 8%. This is the default in WRF-Fire. We did not activate the FMC model in our simulations because it is not activated by default in WRF-Fire. We do activate it in one of our consistency checks in the test directory which confirms the adequacy of our implementation of the FMC model. We added the following: "*The FMC model was not activated in the experiments since this is the default option in WRF-Fire. Instead, FMC was set to its default constant value, 8%.*" and: "*Currently, there are two tests implemented with and without activating the FMC model*".

COMMENT 8

*The issue of horizontal resolution is significant. A resolution of 3 km is inadequate for testing a coupled fire-atmosphere model. This level of resolution causes a dilution of the heat flux across an atmospheric box of that size, leading to under-resolved buoyancy and compromising the two-way coupling effect. Surprisingly, the authors did not reevaluate their choice of horizontal resolution after observing virtually no impact of fire feedback on the atmosphere. For proper validation of a two-way coupled model, it would be beneficial to use in-situ wind observations or airborne vertical velocity observations and plume top height data. Alternatively, model-to-model comparisons could be utilized, provided that the benchmark model is appropriately configured to resolve the processes involved in fire-atmosphere coupling. To effectively resolve fire-atmosphere interactions, coupled models like WRF-Fire are typically run at resolutions of hundreds of meters. For example, WRF-Fire in COFPS operates at approximately 111 m resolution. Consequently, a 3 km resolution is insufficient. If the UFS (Unified Forecast System) does not support a higher resolution, it becomes essential for model developers to create a coupler that can bridge the resolution gap necessary for a coupled fire-atmosphere model. Regarding observational data for model validation, utilizing datasets from experimental campaigns such as FireFlux, CALFIDE, or FIREX-AQ is recommended.*

RESPONSE

We now use 1 km grid spacing in our simulations which is about the maximum grid spacing you can use with UFS at this time. We still see relatively small impacts as a result of activating the feedback from the fire to the atmosphere. However, the impacts of UFS with CFBM are consistent with WRF-Fire (Figs. 9-13) which is the main emphasis in this manuscript. We added a new figure showing impacts in the vertical velocity (Fig. 14), and again we see consistency between the models. We believe it is OK to show a small impact if both models, UFS and WRF, agree, and this is the case.

This is a model to model inter-comparison work and thus our emphasis is not on comparing results against observations, besides the comparison against observed perimeters. Comparing against observations is a logical next step for CFBM now that we are confident in the adequacy of our implementation.

COMMENT 9

*One of the most alarming practices highlighted in the paper is the decision to manipulate the reference height solely to ensure consistency in the simulations. There is no scientific justification for this adjustment. The authors should consider the potential impact this decision may have on future model results, particularly when the wind overestimation in the Unified Forecast System (UFS) is resolved, or in instances where the UFS model accurately captures wind patterns. In such cases, this adjustment could lead to unrealistically low rates of spread (ROS) for fires.*

RESPONSE

We have resolved the wind inconsistencies. It was related to the roughness length which was different in WRF and UFS due to the use of different land surface models. We now use the same parameterizations which resolved the issue. See the updated Figures 7 and 9 and their related descriptions.

COMMENT 10

*Additionally, the rationale for interpolating winds from arbitrary levels to the fire wind height is also unconvincing. The Rothermel model was developed based on laboratory data where the prescribed wind was situated close to the fireline. The experimental data relied on handheld devices deployed at the surface, which were relatively close to the fireline -certainly closer than the described model's horizontal resolution of 3 km. This coupling strategy is integral to WRF-Fire, and altering such a critical component of the model without proper scientific justification is unacceptable. Fire-atmosphere interactions lead to low-level jets, resulting from the inflow into the base of the pyroconvective column, driven by the heat of the fire. This phenomenon is documented, for example, in Benik et al. (2023) https://doi.org/10.3390/fire6090332. Raising the reference height above the layer influenced by the fire contradicts the core principle of coupled fire-atmosphere modeling.*

RESPONSE

This wind interpolation option is from WRF-Fire. We did not change it. Note we did not activate this option in the experiments we presented in the manuscript as described in the experimental set up (Section 4.1).

COMMENT 11

*Another significant shortcoming is the lack of quantitative assessment in the validation section. The paper contains many unsubstantiated and generic statements, such as "The results show consistency between CFBM, UFS, and WRF-Fire, indicating proper implementation and potential for further development." However, the values of the heat fluxes in uncoupled simulations differ by as much as 100% (see Figure 9 at the end of the simulation). Additionally, the simulated fire perimeters are not qualitatively analyzed. It would be advisable to use the Sørensen coefficient for this analysis.*

RESPONSE

We understand the comment about being more quantitative in our evaluation. We now provide quantitative comparisons of the agreement between the simulated fire perimeters. We have calculated the Sorensen coefficient and Heidke skill score and both provide virtually identical results. Figure 1 of this document (Fig. 6 in the new version of the manuscript) shows the Heidke skill score comparing the fire area from UFS-CFBM and WRF-Fire, or the offline CFBM against WRF-Fire. The Heidke skill score is always larger than 0.80 which is large.

We now provide quantitative information regarding the wind speeds. The correlation, bias, and mean absolute error between the wind speed from UFS and WRF is 0.69, -0.3 m/s, and 0.5 m/s, respectively. For the comparison between WRF-1way and CFB-1way the MBE and MAE are 0.1 m/s and 0.2 m/s, respectively; and the correlation is 0.87.

The agreement between the simulated series of the fluxes from WRF and UFS has improved after using the same set of parameterizations, time step and number of vertical layers. We do not see very large spikes now. We also provide quantitative information of the simulated fluxes from the fire. For example, the correlation, bias, and mean absolute error between the heat flux from UFS and WRF is 0.77, 0.15 $10^{-4}$ W/m$^2$, and 1.3 $10^{-4}$ W/m$^2$, respectively. The correlation, bias, and mean absolute error between the moisture flux from UFS and WRF is 0.77, 0.01 $10^{-4}$ W/m$^2$, and 0.12 $10^{-4}$ W/m$^2$, respectively. The correlation is high and the bias and mean absolute error are small compared to the values of the fluxes (see new Fig. 10). Note that the peaks do not need to perfectly align because UFS and WRF use different dynamical cores and the atmosphere is nonlinear which affects the winds and the fire evolution. However, we would expect to see good resemblances in the lower frequency evolution of the fire fluxes from both models, and that is what we see in the new version of the figure (correlation 0.77) with the time series of the fluxes (Fig. 10).

[Figure]

**Figure 1**: Heidke skill score comparing the simulated perimeters from UFS to the WRF-Fire results. The simulation labelled as CFBM is the standalone run driven by WRF data and has been compared against WRF-Fire results

COMMENT 12

*The authors assert that "The CFBM is expected to facilitate joint developments and improve the accuracy of wildland fire simulations, ultimately contributing to better fire management and mitigation strategies." However, the results indicate very poor model performance, and using the uncoupled model still requires WRF and WPS, as the processing tools for the fire data have not yet been developed. This dependency complicates the creation of a community model, given the significant effort needed to build the libraries and set up WRF to generate the WPS.*

RESPONSE

The sentence is just our vision. The main component we highlight in the manuscript is the coupled model. Indeed, we indicate that the standalone model was not the main goal, but it is useful to have. Note that this is a model to model intercomparison to ensure consistency of WRF-Fire and UFS-CFBM and this is what we see in our results. Improving the performance of the model is a logical extension of our work. Herein we highlight the interoperability of the fire model: CFBM is the first fire model available as an Earth system component for coupling with Earth system models via ESMF libraries which are standard for coupling models.

The CFBM relies on the WRF Preprocessing System (WPS) in this first implementation but could be decoupled from WPS. Indeed, our plans are to go beyond WPS in future versions of the model. Although WPS was originally developed for WRF, its reliability, parallelization, and flexibility to manage output variables via namelist or tables make it appropriate for using it for other models. In its simplest version WRF requires Netcdf libraries which is sufficient to compile Geogrid. WRF is also a community model with extended documentation and support. We do not anticipate the compilation of WPS being an issue. Actually, it is already being used by other models. For example, it is used by the Energy Research and Forecasting (ERF, Almgren et al. 2023), and the Model for Prediction Across Scales (MPAS, Skamarock et al. 2012). This last one, MPAS, will be incorporated into UFS and thus WPS will not make an extra requirement for the UFS-CFBM coupling.

Also, relying on WPS in this initial version of the CFBM helps to ensure consistency with WRF-Fire, a major objective in this manuscript, since both models rely on the same preprocessing system. We have clarified in the new version of the manuscript that it is expected that future versions of the CFBM will not rely on WPS: "*In the current version of CFBM, we also rely on the Geogrid program to define the fire grid and interpolate the static datasets, but, since there is no atmosphere component, only the fuels, and elevation (including slopes) in the fire grid are used. It is expected that future versions of CFBM will have its own preprocessing system.*"

COMMENT 13

*Regrettably, this paper cannot be accepted for publication in its current state. While the effort to develop a coupling framework between the fire model and the Unified Forecasting System (UFS) is commendable, the work suffers from significant flaws that fundamentally undermine its scientific validity and contribution to the field.*

RESPONSE

Our emphasis is to check for consistency between UFS-CFBM with WRF-Fire and we now provide an experimental set up of both models that is nearly identical. Hence, the experimental setup is sufficient for testing the consistency of UFS-CFM and WRF-Fire results. The coupling herein presented is the first time a fire behavior model has been made available as an Earth system component available for coupling in Earth system models.

COMMENT 14

*The lack of sufficient innovation in the fire modeling component, poorly designed coupling strategy and numerical experiments, inadequate validation, and poorly organized presentation all contribute to a body of work that does not meet the standards required for publication. Despite the potential value of a robust community fire behavior model and coupling framework, this paper does not provide the necessary advancements or rigor to support such an outcome.*

*This recommendation is not given lightly. Rejecting a paper is always a difficult decision, especially when the effort invested by the authors is evident. However, the issues identified are too substantial to be resolved through even major revisions and require a complete rethinking of the methodology, validation strategies, and presentation. It is hoped that with further refinement, the authors can address these challenges and eventually produce a stronger contribution to the field.*

RESPONSE

The main innovation is that CFBM is the first model available as an Earth system component available for coupling with Earth system models. Not such a model exists. We would not downplay this achievement. We understand this comment as a result of our unclear explanations of the coupling that we have clarified in the new version of the manuscript. The experimental setup is much improved with nearly identical configuration of both models, UFS and WRF, and is sufficient to show the consistency of our implementation with WRF. We have improved the presentation, experimental set up to compare both models, and quantified results.

This is a model description paper and from the aims and scope of the journal website we believe we align well with among other things "*Model description papers are comprehensive descriptions of numerical models which fall within the scope of GMD.*", or "*describe model components and modules, as well as frameworks and utility tools used to build practical modelling systems, such as coupling frameworks or other software toolboxes with a geoscientific application*"; and, again, we are original with our contribution because there is not a fire behavior model available for coupling as a NUOPC component. We believe the manuscript is now suitable for publication. Please, see a more detailed description of the changes introduced in our answers to the following comments.

COMMENT 15

*Specific Comments:*

*L4 "WRF-Fire is a state-of-the-art fire behavior model". Other models like CAWFE or NesoNH-ForeFire offer a similar fictionality. It is unclear what specifically makes WRF-Fire a leading model in this field as suggested by the authors. There are other models built upon WRF that provide a higher level of coupling, including chemistry, as well as more advanced fire parameterizations.*

RESPONSE

We now say the "widely used" model.

COMMENT 16

*L76, L82, 85 The absence of perimeter processing functionality and reliance on WPS are significant shortcomings of a model aspiring to become a community fire model.*

RESPONSE

The perimeter processing is the same as WRF-Fire. This methodology has been successfully used in previous research and thus supported by publications (e.g., DeCastro et al. 2022; Turney et al. 2023). See also answer to comment 7.

Please, see our previous answer to comment 12 for the reliance on WPS.

COMMENT 17

*L105 The current wind interpolation appears to be incorrect. The Rothermel model requires wind speed measurements at mid-flame height. To achieve this, the wind speed should either be interpolated to 20 feet (6.1 meters) and then adjusted using the fuel-specific reduction factors, or it should be directly interpolated to the mid-flame height based on those reduction factors.*

RESPONSE

The explanations are correct, this is how it is done in WRF-Fire. WRF-Fire does not have wind adjustment factors. The user needs to impose the height for the winds that drive the fire evolution. We understand this is something that should be enhanced in the future and we are exploring this possibility (Eghdami et al. 2025). We do say now: "*In the future, we would like to alleviate this subjective choice and implement the use of wind adjustment factors to automatically identify the height that drives the fire evolution based on the fuel characteristics Eghdami et al. (2025).*"

COMMENT 18

*L153 It is unclear where the 2% comes from. Emission factors available in the literature should be used instead. Additionally, the moisture content in the fuel should be accounted for when the moisture fluxes are computed.*

RESPONSE

The fuel moisture content is accounted for in the moisture flux. Please, see the sentence after Equation 5. The 2% is just the default value and follows Coen (2013). We now cite this reference. The factor can be changed in the namelist.

COMMENT 19

*L157 The reorganization of the code should be discussed in more detail. A diagram of the code could be used to support this argument and strengthen this point.*

RESPONSE

We have a diagram illustrating the organization of the code (Figure 1). We have changed the name of the headers to better describe the three categories (the columns in Figure 1). We have also improved our description of the code in Section 2. Please see answers to comments 1, 2 and 3 above. We have also added: "*The code was divided into fire behavior physics, input/output, Earth system model state, driver code, a directory with shared modules, NUOPC coupling code, build infrastructure, and test infrastructure.*"

COMMENT 20

*L163  I recommend using more formal language; instead of "facilitate understanding of what is being done," ,"improve code clarity by relating parts of the code to its physical properties and specific functions".*

RESPONSE

Thanks, we have removed that portion of the text and added a brief sentence saying: "*We have also improved code clarity.*"

COMMENT 21

*L172-175 it would be helpful to explain how this stand-alone code relates to the original stand-alone version of WRF-Fire .*

RESPONSE

This portion of the code is now in Section 2.2.1, the offline coupling Section. After restructuring section 2, it should be clear to understand the standalone, or offline, code. The standalone model was implemented as a way to run the fire behavior model independent of any system. At this time it utilizes existing WRF data to force the fire behavior model and this allows for comparison to the WRF-Fire application, which is the baseline.

We do have in CFBM an idealized version of the code that does not need an atmospheric component and we now show results of this idealized simulations to illustrate the adequacy of our implementation (see new Section 4.2.1).

COMMENT 22

*L175 Change certain to required*

RESPONSE

We have changed this word as suggested.

COMMENT 23

*L182-183 "testing sensitivities in model parameters and methods" consider rephrasing and providing more specific information. For example "testing model sensitivity with respect to XYZ"*

RESPONSE

We now say "*testing sensitivities in model parameters and methods defined in the namelist*".

COMMENT 24

*L183-184 It would be beneficial to provide more details about how the model variables are coupled, the re-mapping strategies used, and so on. The fire code in the examples presented operates at a 100 m resolution, which leads to issues with the handling of fuel and slope data. How is it ensured that the fuel properties are preserved when integrating from a 30 m resolution to a 100 m resolution?*

RESPONSE

We have clarified details about how variables are coupled and the regridding process in the updated Section 2 (see also answer to comment 33).

The interpolation from the fuel/terrain grid spacing to the target 100 m grid spacing is done following methods in Geogrid. There are a number of interpolation options available in Geogrid for doing this and being WRF a community model there is ample documentation about it.

We have added the interpolation methods used to interpolate from the WRF grid to the fire grind: *"The interpolation of variables from the WRF atmospheric data into the fire grid uses a nearest-neighbour interpolation. We plan to include other interpolation methods in future versions of the model."*.

COMMENT 25

*L185 "directories with the fire code", change to directories including the fire code, then modules "used to define"*

RESPONSE

We now say modeling code which is the header of this column in the new Figure 1. We also say *"modules used to define"* now.

COMMENT 26

*L189-190. Rephrase and clarify what "extended with WRF-Fire approaches" means*

*L190 Consider rephrasing to say modules supporting writing standard [...] and reading [...].*

*L193 Consider rephrasing to say "module that supports reading of NetCDF files".*

*L194 Change to module defining…, initializing, updating etc.*

*L194 Instead of "There is also", consider rephrasing to say "the last two modules X and Z are used to decompose the domain into tiles…and …"*

*L200 Consider changing to "when linked to the atmospheric model using two-way coupling".*

RESPONSE

We have clarified all these aspects.

COMMENT 27

*L215. Are these three-dimensional winds – U,V,W or only horizontal winds U,V on the 3D grid? How are winds projected to the terrain plane?*

RESPONSE

Horizontal winds, we have clarified this. Like WRF, the W component is not considered when interpolating to the fire grid.

COMMENT 28

*L217 Capitalize nuopc*

RESPONSE

This is the name of a directory in the code using lowercase. We prefer to use lowercase so it is the same as in the code.

COMMENT 29

*Figure 1 just says nuopc which isn't very informative a higher-level diagram showing modeling components should be used here. Details about the file organization should be presented later.*

RESPONSE

These are the directories in the code. We say this in the caption of Figure 1: "*Each box corresponds with a directory in the code*". In particular, the nuopc directory contains the coupling code. We have clarified this in Figure 1 (the header of column 2 says: coupling code). The explanations of what is inside of the directories are provided in the text. Adding all this information into the figure will result in a figure heavy on text. See the new organization of Section 2. We believe it is clearer now but if not we will be happy to clarify accordingly.

COMMENT 30

*L220 The description of the EXMX is not sufficient. How does it work, is it similar to the WRF Registry mechanism? More explanation is needed here.*

RESPONSE

We have improved our explanations of the ESMX coupling strategy in Section 2: "*The Earth system model connecting the fire and the WRF-Data NUOPC caps that is built for testing uses ESMF or the Earth System modeling eXecutable (ESMX). The ESMF build is the standard, and requires additional code with the NUOPC driver that connects the components, and this code is available in the nuopc directory (Fig. 2). The ESMX application is a community oriented application that builds a NUOPC coupled system, the Earth system model, containing NUOPC compliant components, the CFBM and WRF-Data components in this case, without creating or maintaining the NUOPC driver. Components are added to or removed from an ESMX build through configuration files written in YAML therefore eliminating the need to write code. Hence, using the ESMX extension reduces the amount of code to build the Earth system application, and thus allows for faster developments and easier maintenance. The ESMX build includes an extra test that utilizes an ESMX Data component to prescribe meteorological forcing values. The forcing values are exported to the CFBM which then returns kinematic sensible heat flux and the kinematic latent heat fluxes to the ESMX Data component for validation. These tests, as well as all the offline tests, run nightly to ensure the CFBM and CFBM cap do not break during development. Furthermore, any system utilizing ESMX as its application can embed the CFBM component. Our fire NUOPC component has been an early adopter of the ESMX technology and both coupling strategies, ESMF and ESMX, are being tested.*"

COMMENT 31

*L224 How does "cap" defines the WRF-data grid? Is there a namelist/configuration file that defines variable names, grid location and such?*

RESPONSE

Both the CFBM cap and WRF-Data cap share grid information read from the Geogrid file. This has been clarified.

COMMENT 32

*L226 Figure 1 lists ESMF as a part of the CFBM code, while according to the text says "The main difference is that this offline coupling requires ESMF whereas the standalone code described before does not " This is confusing. Isn't the standalone code using offline coupling? This part needs some clarification.*

RESPONSE

This is a typo and should not contain the word 'offline'. We have removed this word in the new version of the manuscript.

COMMENT 33

*L227 Figure 2 does not effectively illustrate the coupling mechanisms. It is important to differentiate the paths and connections between the modules in offline, online, and standalone scenarios to clarify the data flow and coupling mechanism for each case. Additionally, the coupling between ESMF and ESMX should be explained more clearly.*

RESPONSE

We have clarified our explanations of Figure 2: "*The diagram shown in Figure 2 summarizes the coupling strategy between the Earth system model coupling the WRF-data NUOPC component and the CFBM NUOPC component which is used for testing purposes. NUOPC caps have been written for each model, which allows data to pass from one component to another through NUOPC connectors (arrow). NUOPC connectors are built into NUOPC and redistribute or regrid data between connected pairs of components.*". In order to better explain the offline and online coupling, Section 2 now has 2 subsections, one dealing with the offline coupling (Section 2.2.1) and another with the online coupling (Section 2.2.2). Please refer to the answer to comment 30 for the clarifications between ESMF and ESMX builds.

COMMENT 34

*L239 This line is confusing, did the authors mean that only testing doesn't support MPI, but the rest of the code does support MPI?*

RESPONSE

The CFBM model can be integrated into a MPI parallelized application but at this time the CFBM model runs serially on a single processor. The CFBM model includes MPI library linking for future development. We have added these clarifications to Section 2.3 that deals with the compilation.

COMMENT 35

*Organizing Figured 2 and 3 as a two-panel plot to highlight the similarities and differences between the coupling strategies would be beneficial.*

RESPONSE

We considered joining the figures. In the end we kept them separately because they belong to different sections of the manuscript. Figure 2 is described in Section 2 and Figure 3 in Section 3. We have added some text to describe similarities and differences. Similarities being that a single NUOPC cap written for CFBM is agnostic to the external atmosphere being coupled. Differences being that a completely different atmospheric component, whether data or not, can be coupled therefore creating an entirely different coupled Earth system model. These explanations were included in Section 3.

COMMENT 36

*L243. How is the machine environment specified? What information is included there? How would users add other machines?*

RESPONSE

To ensure reproducibility and ease of use the environment used to build executables and run tests can be automatically configured, via adding --env-auto to the compile script. This flag then locates a bash script that loads modules and defines environment variables for defined systems. At this time the only defined system is 'derecho'. New systems can be defined by adding a <system_name> directory containing bash configuration files. The new system must then be added to the 'auto environment' section of the compile script. We have added these explanations to Section 2.3.

COMMENT 37

*L253, Where is the fire namelist? Shouldn't it be listed in Figure 1?*

RESPONSE

Yes, but we do not want to go deeper in the directory structure. There are examples inside the tests directory. We now say this in the text: "*Examples of the namelists are provided in the tests directory, inside the directories for each test.*"

COMMENT 39

*L258-259 Is there any specific reason why the original coupling strategy from WRF with the extinction depth was abandoned? How would the model be kept stable under rapid fire growth and an instantaneous initialization of the fire from perimeters?*

RESPONSE

It makes sense to release the heat from the ground fire in the first model layer, like the land-surface model fluxes. And we did not see any instabilities with this UFS implementation. It is possible that when UFS allows for a finer grid spacing we have to revisit this implementation, but should not be a concern at this moment.

COMMENT 40

*L273-274. This sentence needs rephrasing. It is unclear how "WRF-Fire procedures" are related to the comparison between the results from the models.*

RESPONSE

Our implementation of CFBM closely follows WRF-Fire methods as described in Section 2.1. We now point to Section 2.1 in the sentence: "*The version 0.2.0 of the CFBM herein presented closely follows WRF-Fire methods (see description in Section 2.1) in order to compare results between the CFBM NUOPC component implemented in UFS and WRF-Fire results.*"

COMMENT 41

*L274. More justification is needed for selecting this specific fire for comparison. Given the poor performance of WRF-Fire in this fire event, this choice is confusing.*

RESPONSE

We selected this fire because it is the largest wildfire perimeter in Colorado's history. We say this in the manuscript. Our main emphasis in this manuscript is to show consistency between WRF-Fire and UFS-CFBM and for this purpose this case is sufficient. Hence, our main emphasis is in the model intercomparison and this is indicated in the title of the section 4: "4 Model inter-comparison: …".

COMMENT 42

*L278. Considering the wide selection of parameterizations in WRF it should be possible to reduce the setup differences to the dynamics core alone. Such a comparison would be much more meaningful.*

RESPONSE

Yes, the new version of the manuscript now uses the same set of parameterizations in UFS and WRF.

COMMENT 43

*L280 A wind-driven fire like the one described here is not the best test case for a coupled fire-atmosphere model. Under strong wind conditions, evaluating the role of fire-atmosphere coupling processes is challenging.*

RESPONSE

A wind driven case is good enough for our purposes of model intercomparison. If something is wrong in the coupling of the winds from the atmosphere with the fire it should be very evident. And it is also sufficient to compare the magnitude of the fire fluxes and the impact in the atmosphere which is what we show in the manuscript (Figs 9-14).

COMMENT 44

*L293. This is a significant modification. Are there any observational data supporting the need for that? Do experimental data from experiments like FireFlux and FireFlux II suggest that the wind profile before and after changed significantly due to the roughness change?*

RESPONSE

The WRF roughness length changed from the first time step to the next one and then remained constant. WRF-Fire used the first roughness length only. With this change we ensure the atmosphere and the fire "see" the same roughness length all the time steps. It is a change to ensure model consistency. We now say: "*First, we update the atmospheric roughness length used by the fire model every time step since it varies between the first time step and subsequent ones.*"

COMMENT 45

*How was the roughness change estimated? Was it modified in both models?*

RESPONSE

Yes, both models, WRF-Fire and CFBM in standalone mode (running offline reading WRF data) use consistent roughness lengths. Please see the answer to the previous comment.

COMMENT 46

*Correct the tense "we updated"*

*Consistency in the tense. The text is hard to follow and should be rephrased.*

*L302 Consider changing to "Consistency between simulations"*

RESPONSE

We now say we "fixed" the Table, and indicate this is a fix in the official WRF release. We changed the old line 302 as suggested.

COMMENT 47

*L305- The ignition procedure requires a more detailed description, especially regarding the coupled simulations. Sudden ignition can disrupt the interaction between fire and atmospheric processes, leading to poorer model results. Greater care must be taken to ensure that the integration of fire perimeters is conducted in a manner that keeps both the fire and atmospheric model components in sync at the start of the simulation.*

RESPONSE

The simulations starting from a fire perimeter are one way simulations to compare with the standalone version of the code which only runs in one way. Results for the coupled simulations are shown in the section dealing with the point ignitions.

COMMENT 48

*L320 It is puzzling why the WRF time step and the number of levels have not been simply adjusted to match those in UFS. The study would benefit from a more careful experimental design.*

RESPONSE

We now use the same number of vertical layers and the same time step in both models.

COMMENT 49

*L327. Why 20 minutes? This creates inconsistencies between the uncoupled WRF simulation and the new implementation. Are the wind variables averaged over the 20 minutes or are instantaneous values used?*

RESPONSE

We saved WRF output, which is always instantaneous, every 20 min. We understand that having more frequent outputs should provide better agreement between WRF-Fire and our offline simulations with CFBM, but the agreement is good enough to suggest the adequacy of our implementation.  For the comparison between the wind speed from WRF-1way and CFB-1way the MBE and MAE are 0.1 m/s and 0.2 m/s, respectively; and the correlation is 0.87. In anycase, our main emphasis is agreement between UFS-CFBM and WRF-Fire and in these simulations the atmospheric state is updated every time step. And again, we see consistency of the results from both models.

COMMENT 50

*L330- What is the scientific basis to believe that at 3km a coupled fire-atmosphere model could successfully resolve fire-atmosphere interactions? It is over an order of magnitude coarser mesh than COFPS, or other fire studies using coupled models.*

RESPONSE

We now use 1 km grid spacing which is on the limit of what can be done with UFS. Other coupled fire-atmosphere modelers have used similar or coarser grid spacings (e.g., Kochanski et al. 2019, Michael et al. 2022). The 1 km grid spacing is fine enough to test the consistency between UFS-CFBM and WRF-Fire. This is demonstrated with the consistency in the time series of the fluxes and the impacts in the atmospheric variables that albeit small are sufficient to compare WRF-Fire and UFS-CFBM (Figs. 9-14).

COMMENT 51

*L335 It must be explained what the purpose of running a coupled model at such low resolution is, and what exactly the authors expected to achieve by running simulations in such a configuration.*

RESPONSE

We now use 1 km grid spacing which is sufficient to test the consistency between WRF-Fire and UFS-CFBM: *"However, the impacts are sufficient to assess the consistency between WRF and UFS fire simulations which is the main objective of this experiment."*. See also answer to the previous comment.

COMMENT 52

*L338 It is unclear based on what the authors made this decision. Rothermel internally uses wind speed at the mid-flame height. The midflame winds are specified using reduction factors that are used to convert winds from 6.1 (20ft winds) to the midflame height. Such changes invalidate the simulations.*

RESPONSE

We closely follow WRF-Fire methods in the manuscript. WRF-Fire does not have wind adjustment factors. The user needs to impose the height for the winds that drive the fire evolution. We understand this is something that should be enhanced in the future and we are exploring this possibility (Eghdami et al. 2025). We do say now: "*In the future, we would like to alleviate this subjective choice and implement the use of wind adjustment factors to automatically identify the height that drives the fire evolution based on the fuel characteristics Eghdami et al. (2025).*"

COMMENT 53

*L350. That is not true. There is no quantitative analysis so this claim is subjective. Also, the fact that the authors had to adjust the fire wind height from 5 to 2.5 meters to get there is alarming.*

RESPONSE

We now have a new Figure (Figure 6, and Fig. 1 above) that shows the Heidke skill score comparing UFS against WRF-Fire. The Heidke skill score is around 0.9, and always larger than 0.8 (perfect agreement is 1.0), which suggests consistency of the simulations. We now use the same wind height in the simulations.

COMMENT 54

*L351. The term "underestimate" is not appropriate in this context. The simulation is completely unrealistic, likely due to a configuration error in the WRF-Fire model, which then affects the proposed model. Additionally, without proper validation of the basic weather variables, it is impossible to determine whether the issue lies with the fire or the weather aspects of the model. Overall, it is evident that both models have failed, but this does not provide any conclusive evidence regarding the implementation of the model.*

RESPONSE

We have redone the simulations at 1 km grid spacing. There is no configuration error, the simulations underestimate the fire growth. We show the observations of the perimeter as a reference since our emphasis is on the model intercomparison. Hence, we do not focus on the comparison with observations, we focus on comparing WRF-Fire and UFS-CFBM. We now provide a quantitative comparison of the simulated perimeters, the Heidke skill score (Figure 6, and Figure 1 in this document), which shows large values, around 0.9, when comparing the CFBM simulations with WRF-Fire pointing to adequate behavior of CFBM with respect to WRF-Fire.

COMMENT 55

*L358 This kind of discrepancy should be minimized through careful planning of the model setup. Running a standalone Fire code with UFS forcing would better serve the purpose of this experiment.*

RESPONSE

We now use a very close set up of the models with the same parameterizations, same time step, and same number of vertical levels. We have updated this description of the new Figure 7. It is not possible to run the

standalone fire code using UFS at this point. The standalone code uses WRF data to compare it with WRF-Fire to ensure we are able to reproduce WRF-Fire results with CFBM.

**COMMENT 56**

*The mean wind speed varies by as much as 1 m/s, which represents a significant difference of 20-25%.*

RESPONSE

We have updated the figure comparing the wind speeds since we  now use the same height in both models, and a very similar model set up. There is good agreement now, and most of the time the differences are below 1 m/s. See the quantification in the answer to comment 11. There are differences between UFS and WRF because they are different models with different dynamical cores, but we would expect an overall good agreement as shown in the figure.

**COMMENT 57**

*On a "hot, dry, and windy day," as mentioned in L281, why are the wind speeds only around 4 m/s, which is less than 9 mph? Previously, gusts were noted to reach 71 mph.*

RESPONSE

This is the wind speed near the ground, at 2.5 m above ground level. The wind speed at the first model level, around 24 m above ground level is larger than 6 m/s in both models.

**COMMENT 58**

*The difference between the CFB and UFS-1-way models is alarmingly high, especially considering their resolution.*

RESPONSE

There is better agreement in the winds in the new set of runs using 1 km grid spacing and the same parameterizations, time step, and number of vertical layers. The correlation, bias, and mean absolute error between the wind speed from CFB and UFS-1-way is 0.74, 0.4 m/s, and 0.5 m/s, respectively. Again, differences are expected because UFS and WRF use different dynamical cores.

**COMMENT 59**

*L361 The "consistency" should be quantified. The plot suggests something opposite. There is no consistency – there is no systematic bias. The WRF winds are higher or lower than the CFB or UFS-1-way depending on the time.*

RESPONSE

We have updated the figure with results from the new simulations. We now quantify the agreement between the simulations to provide a better understanding of the consistency between the models. The correlation, bias, and mean absolute error between the wind speed from UFS and WRF is 0.69, -0.3 m/s, and 0.5 m/s, respectively.

For the comparison between WRF-1way and CFB-1way the MBE and MAE are 0.1 m/s and 0.2 m/s, respectively; and the correlation is 0.87. There is sufficiently good consistency now.

COMMENT 60

*L365-366. "As has been already mentioned, to obtain consistency in the simulated winds from WRF and UFS we had to use a different height for the winds driven the fire in the models". This is unacceptable, especially because after the adjustment between 8/14/ 03 to 8/14/13 UFS showed lower values than WRF. Change "driven" to "driving"*

RESPONSE

We now use the same height, 2.5 m, in both models.

COMMENT 61

*Figure 5. The actual wind conditions at the same altitude should be presented. Additionally, validation against observed data is necessary. If the Unified Forecast System (UFS) is unable to realistically simulate winds and requires such corrections, its suitability as a platform for a community fire model is highly questionable.*

RESPONSE

We now show the winds at the same height. It is not our main focus to evaluate the realism of the simulation but the consistency between UFS-CFBM and WRF-Fire. This manuscript focuses on a model inter comparison. In the new experiments, the correlation, bias, and mean absolute error between the wind speed from UFS and WRF is 0.69, -0.3 m/s, and 0.5 m/s, respectively. This is good consistency considering UFS and WRF are different models. We now provide the quantification of the agreement between the wind speed simulated by UFS versus WRF.

COMMENT 62

*Figure 6. Explain how the fire front is defined here, based on the heat flux threshold, fuel fraction, air temperature?*

RESPONSE

The fire front is defined as the grid cells that are burning. We no longer have this figure in the manuscript but we have added the description in the caption of Figure 7 that also shows time series at the fire front.

COMMENT 63

*L372. The consistency hasn't been quantified or analyzed yet. At that point, we only know that after creating an artificial adjustment to the wind height, the results are comparable. For example, we don't know if the fuel moisture is consistent, if the fluxes at the fire mesh are comparable, whether fuel consumption is comparable, etc.*

RESPONSE

Now we use the same height to calculate the winds that drive the fire evolution. The results are comparable as well, and we now provide quantifications of the agreement between UFS-CFBM and WRF-Fire using the Heidke skill score. See new Figure 6 (Fig. 1 in this document). The skill score is around 0.9, and always larger than 0.8, being 1.0 a perfect comparison. The fuel moisture content is set to the default values, 8%, and we now say this in the new version of the manuscript. In one-way simulations (Section 4.2.2) there are no heat and moisture fluxes. These are ultimately a function of the fuel consumption and are analyzed in this section (Section 4.2.3) dealing with 2-way simulations. We have clarified the opening sentence: "*After ensuring consistency of the perimeter evolution in one-way coupled simulations, we turn our attention to the two-way coupled experiments (WRF-2way and UFS-2way) that also allow for inspecting fire fluxes and atmospheric impacts.*"

COMMENT 64

*L374. The claim "Again, we see an underestimation of the observed burned area with consistency between the simulated perimeters using WRF and UFS" is not supported.*

RESPONSE

The underestimation is supported with the results shown in the figure. The consistency of the perimeters is also shown in the figure and now we have the quantification of the agreement of the simulated parameters calculated with the Heidke skill score shown in Figure 6 (Fig. 1 above). We now indicate that the Heidke skill score is around 0.9.

COMMENT 65

*There is no quantitative analysis to support this claim. Interestingly, the backfire rate of spread (ROS) differs significantly between the models, with the WRF model showing a more rapid backfire propagation compared to the UFS. Since the backfire ROS is essentially derived from the no-fire, no-wind ROS from the Rothermel model, one would expect them to match. These discrepancies should not be affected by the modeled winds.*

RESPONSE

The backfire rate of spread is the same in WRF-Fire and CFBM. What the reviewer indicates is backfire, it is actually not backfire but the winds moving forward the fire front in this part of the perimeter.

COMMENT 66

*L375-376 "This is the first evidence of a relative small impact of the heat and moisture fluxes from the fire for these experiments that use 3 km grid spacing." Unfortunately, this is also evidence that the modeling experiment wasn't carefully planned and is unsuitable for validating coupled fire-atmosphere models.*

*It should also say here and in a few other places "relatively" not "relative".*

RESPONSE

We now use 1 km grid spacing and the experimental setup is sufficient to test the consistency between UFS-CFBM and WRF-Fire. There may be a small impact in the atmosphere but both UFS-CFBM and

WRF-Fire are consistent in the magnitude of the fire fluxes (Fig. 10) and the impacts in the atmospheric variables (Figs. 11, 12, 13 and 14).

We have changed "relative" for "relatively", thanks for pointing this out.

COMMENT 67

*L376 Rephrase "In this direction, we further show the evolution of the burnt area and the wind speeds calculated with the 1way and 2way experiments using WRF and UFS (Fig. 8)"*

RESPONSE

We have improved the sentence.

COMMENT 68

*L380. "The consistency is also evident in the time series of the fire heat and moisture fluxes and the smoke emissions (Fig. 9)." The plots actually show something different. The fluxes from on-way simulations differ by as much as 50% on 08/14-03, or 08/14-05. At the end of the simulation, the WRF-1way heat fluxes are twice as high as the ones from the UFS-1way run. In the two-way simulations, we see identical heat fluxes with UFS-1way and UFS-2way, which is surprising considering that on 08-14 12 the couped winds are visibly weaker than uncoupled. A careful explanation of these issues is needed.*

RESPONSE

This Figure is new since we now use 1 km grid spacing, the same parameterizations, time step, and number of vertical layers. There is an overall good agreement with the evolution of the fluxes (correlation 0.77) with some alternating peaks in the models that do not necessarily coincide in time. This is ultimately related to differences in the winds which is expected considering UFS and WRF use different dynamical cores. We have clarified these aspects in the new version of the manuscript. We also provide quantification of the agreement between the times series based on the correlation, bias and mean absolute error (see answer to comment 11).

In the new simulations the fluxes are similar but not identical, like the wind speed.

COMMENT 69

*If the time step was an issue, the data should be time aggregated to the same intervals. However, I'm not convinced that WRF heat fluxes are averaged in time. More explanation is needed about why the instability appears at the ignition time. See also the comment about the ignition procedure. At what rate is the ignition implemented? The ignition needs a more thorough explanation and more careful planning.*

RESPONSE

We have redone the simulations using the same time step and we do not see the large spikes in UFS. There are some alternating peaks. Note that the peaks do not need to perfectly align because UFS and WRF use different dynamical cores and the atmosphere is nonlinear which affects the winds and the fire evolution. However, we would expect to see good resemblances in the lower frequency evolution of the fire fluxes from both models, and that is what we see in the new version of the figure (correlation 0.77) with the time series of the fluxes (Fig.

10). Note that these are point ignition simulations. Both models are consistent in the ignition process and we now see consistency in the fluxes (See new Fig. 10).

**COMMENT 70**

*Figures 10 and 11. The header should be changed. It is unclear what "WRF-Fire x UFS-Fire" means. It suggests a multiplication between the two, which is misleading.*

RESPONSE

Yes, we agree, we have changed the title according to this comment.

**COMMENT 71**

*L386. If the timestep is to blame why didn't the authors run WRF with the same time step? This is again the problem of inconsistency between the model configuration that could have been avoided if the numerical experiments were carefully planned.*

RESPONSE

We now use the same time steps and we have resolved the problem of the large spikes. We have removed the part describing the issues related to using different time steps.

**COMMENT 72**

*L393. "The WRF differences are positive because WRF has a dedicated variable for the fire fluxes in the atmospheric grid and the fact that the fluxes are zero in the one-way experiment." Actually, WRF also has an integrated variable GRNHFX that should be used here to keep it consistent with UFS since it provides only aggregated fluxes. Alternatively, a new diagnostic variable for fire heat fluxes could be added to UFS.*

RESPONSE

We were not clear with this sentence. We wanted to say that WRF has a dedicated variable for the fire flux but UFS does not. The fire flux is incorporated into the surface flux variable in UFS. It is not as easy as in WRF to add variables into UFS. We have clarified the sentence.

**COMMENT 73**

*L395. This claim is not supported by the presented results. The UFS simulation, particularly in the bottom right of Figure 10, shows a hot plume downwind from the fire, which is absent in the WRF simulation. This discrepancy should be investigated further. Additionally, a comparison of the overall heat fluxes from WRF and UFS displayed as the difference (WRF - UFS) would provide more informative insights. The same applies to temperature comparisons.*

RESPONSE

This figure shows the fire heat flux from WRF and the fire heat flux plus the surface heat flux from the land surface model for UFS (see answer to previous comment). The fire signal in the heat flux is only valid at the

location of the fire as we indicate in the text. The discrepancies mentioned are outside of the simulated fire perimeter valid for 12 h into the simulation. The discrepancies are going to grow due to nonlinearities in the atmospheric evolution, that is why the plots at 3 h and 6 h into the simulation look better. It is true that the UFS fire heat flux is more spread across atmospheric grid cells than WRF. We have clarified the sentence. We now say "*However, at the location of the fire (Fig. 7) both models show positive differences of similar magnitude, with the UFS impact spread across more grid cells, at the three times shown.*". For the case of temperature, we will expect the impact from the fire to be in better agreement between the models at the beginning of the simulation before the nonlinear effects grow and this is why we see better agreement 3 h and 6 h into the simulation that at 12 h into the simulation.

**COMMENT 74**

*What is the size of the domains used for testing? Why are the data shown on a coarse 21x17 mesh? Is it the native atmospheric mesh?*

RESPONSE

The atmospheric grid in WRF has 629 by 599 grid points and the grid in UFS has 599 by 570 grid points. We have added this information to the manuscript. We show the atmospheric grid around a small region around the fire. We have incorporated the size of the atmospheric grid in the new version of the manuscript.

**COMMENT 75**

*L405 The change in nomenclature is confusing. The switch from UFS-1 (one-way coupled) and two-way coupled UFS and CFB makes the plots difficult to analyze. Additionally, the lack of axes and labels in Figures 7 and 10-13 is unacceptable, especially since a figure depicting the domain configuration is missing.*

RESPONSE

The nomenclature was introduced in the section with the experimental setup (Section 4.1) and the label of the experiments is shown in Table 3. We have put axis with the geolocation in the new version of Figure 7, now Figure 8, that allows one to geolocate the location of the fire. This helps with the geolocation of the new version of Figures 10-13 (now Figures 11-13) since these figures include the observed fire perimeter like in Figure 8.

**COMMENT 76**

*L406 The sentence, "This spread is simulated accurately in the northern and eastern parts of the fire, but results in a clear overestimation towards the northeastern portion of the perimeter," needs rephrasing as it is self-contradictory. It suggests that the northeastern progression is both accurate and overestimated. The reality is that the progression does not appear to be particularly good. This is particularly concerning in light of findings from the ignition point analysis, which suggested a significant underestimation of fire growth, while now we see a notable overestimation. I recommend that the authors review their approach to representing fuel moisture. The fact that the model overpredicts the rate of spread (ROS) at 10:00 in the morning but performs considerably better at 16:30, when fuel moisture reaches its minimum, indicates that fluctuations in fuel moisture are not represented accurately. Proper conditioning and modeling of fuel moisture are critical for analyzing diurnal fire activity.*

RESPONSE

We have rewritten these portions of the manuscript to describe results from the new simulations. There is a remarkable good agreement between the modeled perimeters which is quantified in terms of the Heidke skill score (New Figure 6, and Fig. 1 above). We do acknowledge limitations with respect to observations but highlight the agreement between UFS-CFBM and WRF-Fire which is our main objective here to test the adequacy of our implementation of the fire model in UFS. After ensuring consistency between the models, our next logical step is a more detailed verification with observations in future studies.

COMMENT 77

*L407 The issue of representing fire activity based on infrared perimeters has been previously studied. A potential solution to this problem is discussed in: https://doi.org/10.3389/ffgc.2023.1203578. The authors should consider employing such a method to enhance the realism of their simulation.*

RESPONSE

Thanks for the suggestion, we will keep it in mind for subsequent studies where we will focus on comparing CFBM with observations. Herein we focus on the model inter-comparison.

COMMENT 78

*L411 The differences between UFS and CFB are significant, and it is crucial to investigate why uncoupled simulations show faster progression. Generally, fire-induced winds tend to accelerate the fire, so we would expect the opposite effect. If the variations in the frequency of inter-model communication result in such significant differences, the proposed system cannot be considered robust.*

RESPONSE

The simulations starting from fire perimeters are all one-way simulations since the standalone CFB can only run in one way. We have redone the simulations with the same parameterizations, time step, and number of vertical layers, and now there is a much better agreement (Figure 14 in the revised version of the manuscript). The Heidke skill score comparing the agreement between UFS and WRF perimeters for the simulations starting from an observed fire perimeter is high, around 0.95. There is consistency in the simulated perimeters.

COMMENT 79

*L420 The simulated wind should be validated to provide more insight into these differences. Wind speed alone is not sufficient; wind direction is also critical. The authors should consider investigating this aspect as part of their analysis.*

RESPONSE

The focus of the manuscript is model intercomparison. We now provide a quantification of the agreement of the simulated perimeters using the Heidke skill score. A comparison with observations is our next logical step after we have confirmed the adequacy of our implementation of the fire model.

COMMENT 80

*L421-423 The plots indicate the opposite: CFB systematically overestimates fire expansion compared to UFS, which also consistently overestimates fire growth along the active parts of the fire. The runs are only consistent regarding backfire propagation and propagation along the northern and northwestern flanks.*

RESPONSE

We have reformulated our explanations because we now have a remarkably good agreement between the models for the simulation starting from perimeter number 2. Again, we emphasize the emphasis in the model intercomparison that was not clear enough in the previous version of the manuscript.

COMMENT 81

*L439 Refer to comment L4.*

RESPONSE

We now say the "widely used" model.

COMMENT 82

*L446  The work presented contradicts this statement, as it highlights significant inconsistencies between WRF and CFBM. There is underestimation when ignited from a point and overestimation when ignited from a perimeter.*

RESPONSE

There is consistency between the simulated perimeters and this is now quantified using the Heidke skill score (Fig. 6, and Fig. 1 above). The Heidke skill score is around 0.9 and always larger than 0.8. We have added a reference to the new figure showing the Heidke skill score to support this statement.

COMMENT 83

*L448 Unfortunately, this statement lacks substantiation. Fundamentally, it is impossible to evaluate coupled models using numerical experiments that are inadequate to resolve the processes that these models intend to represent.*

RESPONSE

We do see a small impact of the fire feedbacks, but the feedbacks are consistent between UFS and WRF. This is exemplified in the manuscript by the comparison of the time series of the fluxes from the fire, as well as spatial plots with the heat flux that goes into the atmosphere, impacts in temperature and in moisture (see the updated Figures 9-13 and the new Fig. 14).

COMMENT 84

*L454 The fire community would greatly benefit from a more modular construction of fire models, and greater attention should be given to this aspect.*

RESPONSE

CFBM is designed with modules having orthogonal code in order to facilitate maintenance and future extensions, and maximize the interoperability with different  atmospheric models. We have highlighted the interoperability part in the new version of the manuscript.

**References**

Almgren et al., 2023: ERF: Energy Research and Forecasting. Journal of Open Source Software, 8(87), 5202, https://doi.org/10.21105/joss.05202

Coen, J., 2013: Modeling Wildland Fires: A Description of the Coupled Atmosphere-Wildland Fire Environment Model (CAWFE), Tech. rep., NCAR Technical Note NCAR/TN-500+STR., Boulder, CO.

DeCastro, A. L., T. W. Juliano, B. Kosovic, H. Ebrahimian, J.K. Balch, 2022: A Computationally Efficient Method for Updating Fuel Inputs forWildfire Behavior Models Using Sentinel Imagery and Random Forest Classification. Remote Sens., 14, 1447. https://doi.org/10.3390/rs14061447

Eghdami, M., P. A. Jimenez y Munoz, and A. DeCastro, 2025: Sensitivity to the representation of wind for wildfire rate of spread: Case studies with the Community Fire Behavior model. Fire, 8, 135.

Kochanski, A.K., D.V. Mallia, M.G. Fearon, J. Mandel, A.H. Souri, and T. Brown, 2019: Modeling wildfire smoke feedback mechanisms using a coupled fire-atmosphere model with a radiatively active aerosol scheme. Journal of Geophysical Research: Atmospheres, 124, 9099-9116.

Kochanski, A. K., K. Clough, A. Farguell, D.V. Mallia, J. Mandel, and K. Hilburn, 2023: Analysis of methods for assimilating fire perimeters into a coupled fire-atmosphere model.  Front. For. Glob. Change 6:1203578. doi: 10.3389/ffgc.2023.1203578

Michael, Y., G. Kozokaro, S. Brenner, and I.M. Lensky, 2022: Improving WRF-Fire wildfire simulation accuracy using SAR and time series of satellite-based vegetation indices. Remote Sensing, 14, 2941.

Skamarock, W. C., Klemp, J. B., Duda, M. G., Fowler, L. D., Park, S.-H., & Ringler, T. D., 2012: A multiscale nonhydrostatic atmospheric model using centroidal Voronoi tessellations and C-grid staggering. Monthly Weather Review, 140, 3090–3105.

---

## Author Comment (AC3)

"The Community Fire Behavior Model for coupled fire-atmosphere modeling: Implementation in the Unified Forecast System" by Pedro A. Jiménez y Muñoz, Maria Frediani, Masih Eghdami, Daniel Rosen, Michael Kavulich, and Timothy W. Juliano.

—--------------------------------------------------------------------

Reviewer #3

*Review for "The Community Fire Behavior Model for coupled fire-atmosphere modeling: Implementation in the Unified Forecast System" The manuscript introduces the Community Fire Behavior Model (CFBM), a newly developed fire behavior model designed for seamless coupling with different atmospheric models using the Earth System Modeling Framework (ESMF). The key objective of CFBM is to provide a flexible and modular framework for simulating coupled fire-atmosphere interactions without requiring model-specific interpolation procedures. This approach is intended to foster broader collaborations and model integrations beyond the traditional Weather Research and Forecasting (WRF) model with fire extensions (WRF-Fire). The current review starts with specific comments in each section of the manuscript and finishes with a general comments section, followed by a final decision section.*

RESPONSE

We would like to thank this reviewer for the time she/he devoted to reviewing the manuscript and for the comments provided that have helped to increase the quality of the manuscript. Below we reproduce the comments, followed by a detailed answer that includes the changes introduced in the manuscript. These include, among other things, running new simulations with a more similar setting of WRF-Fire and UFS-CFBM (same set of parameterizations, time step, and number of vertical layers), adding quantifications of the agreement between the simulations, adding an idealized case study, and checking for consistency in the simulations in sensitivity experiments. We believe the manuscript should be appropriate for publication this time.

COMMENT 1

*The fire behavior model:*
*The input data interface remains dependent on Geogrid from WPS, which necessitates pre-processing via WRF. Consider addressing whether the model can be decoupled from WRF-specific tools.*

RESPONSE

The CFBM relies on  the WRF Preprocessing System (WPS) in this first implementation but could be decoupled from WPS. Indeed, our plans are to go beyond WPS in future versions of the model. Although WPS was originally developed for WRF, its reliability, parallelization, and flexibility to manage output variables via namelist or tables make it appropriate for using it for other models. Actually, it is already

being used by other models. For example, it is used by the Energy Research and Forecasting (ERF, Almgren et al. 2023), and the Model for Prediction Across Scales (MPAS, Skamarock et al. 2012). This last one, MPAS, will be incorporated into UFS and thus WPS will not make an extra requirement for the UFS-CFBM coupling.

Also, relying on WPS in this initial version of the CFBM helps to ensure consistency with WRF-Fire, a major objective in this manuscript, since both models rely on the same preprocessing system. We have clarified in the new version of the manuscript that it is expected that future versions of the CFBM will not rely on WPS "*In the current version of CFBM, we also rely on the Geogrid program to define the fire grid and interpolate the static datasets, but, since there is no atmosphere component, only the fuels, and elevation (including slopes) in the fire grid are used. It is expected that future versions of CFBM will have its own preprocessing system.*"

COMMENT 2

*The results could benefit from improvements to the fire initialization since the method presented seems to mismatch fire and atmosphere dynamics. The authors should consider using a spin-up period during the perimeter initialization.*

RESPONSE

We use the same fire initialization procedures in UFS-CFBM and WRF-Fire to check for consistency between results of both fire behavior models. The initialization procedure we use is the only option available in WRF-Fire. This methodology has been successfully used in previous research and thus supported by publications (e.g., DeCastro et al. 2022; Turney et al. 2023). We understand there are other initialization methods that can be explored in future versions of the CFBM (e.g., Kochanski et al. 2023) where our intention will be to go beyond current WRF-Fire methods. We clarified these aspects in the new version of the manuscript when describing the fire initialization methods: "*This fire-perimeter initialization method is the only available in WRF-Fire and has been successfully used in previous research with the WRF-Fire model (DeCastro et al. 2022; Turney et al. 2023). More sophisticated methods (Kochanski et al. 2023) could be explored in future versions of the model that will go beyond WRF-Fire methods.*"

The simulations we present in the paper starting from an observed perimeter are one way simulations in order to compare with the simulation we performed with the standalone version of CFBM that reads WRF data offline. The fire and atmospheric dynamics do not play a role here. This experiment aims to check for the consistency of UFS and WRF-Fire when starting from a given fire perimeter, and our results show this consistency.

The perimeter initialization runs have a spin-up period. This information was missing in the previous version of the manuscript. We have added the information: "*This leaves 2.7 h of spin up for the Perimeter-1 runs and 0.9 h for the Perimeter-2 runs*".

COMMENT 3

*The two available options for fire wind interpolation require further explanation. Additionally, if wind reduction factors are applied, their implementation and rationale should be discussed.*

RESPONSE

We have improved our explanations: "*Two interpolation options are available. The first one uses a linear interpolation on the logarithm of height from two adjacent model layers to the target height. In the second one, the user specifies a second height and the interpolation is performed from this height to the target height using the logarithmic wind profile.*". These are the options available in WRF-Fire. There are no wind adjustment factors (WAF) for the moment in CFBM since we follow WRF-Fire methods in our current implementation. We are currently exploring the option of adding WAFs (Eghdami et al., 2025). We have clarified this too: *In the future, we would like to alleviate this subjective choice and implement the use of wind adjustment factors to automatically identify the height that drives the fire evolution based on the fuel characteristics Eghdami et al. (2025).*"

COMMENT 4

*The manuscript should specify whether Lambert-Conformal projection is the only supported coordinate system in CFBM. If additional projections are supported, this should be clarified.*

RESPONSE

Yes, we agree, we have clarified this aspect: "*the WRF projection must be Lambert-Conformal in CFBM version 0.2.0. Support for Mercator and polar stereographic projections will be added in future versions.*".

COMMENT 5

*Experimental setup:*
*The downscaling from a 3 km atmospheric grid to a 100 m fire mesh presents notable limitations:*
*- The atmospheric resolution is too coarse to adequately capture fire atmosphere interactions.*
*- The fire mesh resolution is also too coarse to properly resolve fire spread parameterizations.*

RESPONSE

We used 3 km grid spacing for the atmospheric grid spacing because it is the one used in the UFS-based Rapid Refresh Forecast System (RRFS) but in the revised version of the manuscript we use 1 km grid spacing. This largely reduced the ratio of the atmosphere to fire grid size (10 in the current version of the

manuscript). Going beyond this atmospheric grid spacing cannot be easily justified in UFS since we start to go into the "gray zone" or "terra incognita" region where is not clear how to parameterize turbulence effects; and it is not possible to use a large-eddy simulation (LES) approach in UFS currently (it is possible in WRF). Hence, we stay at 1 km grid spacing because it is at the limit of what is possible with UFS and it allows us to have similar model configurations to evaluate the consistency between UFS-CFBM and WRF-Fire which is the main objective of the manuscript. The feedback that we see from the fire to the atmosphere, although small, is consistent between UFS-CFBM and WRF-Fire (Fig. 10 with the time series of the fluxes, and Figs. 11-14 with the feedback to the atmosphere). Hence, the configuration is sufficient to support the conclusions of the manuscript.

We have incorporated the previous rationale in the new version of the manuscript: "*The simulations were configured with a single domain covering the state of Colorado at 1 km horizontal grid spacing. The WRF domain has 629 by 599 grid points whereas the UFS domain has 599 by 570 grid points. Although going to finer grid spacing is desirable to better represent fire-atmosphere interactions using a turbulence resolving mode based on large-eddy simulation (LES), this is not possible with UFS that currently lack a LES approach. Hence, we used 1 km because is at the limit of what can be achieved with traditional planetary boundary layer parameterizations available in both UFS and WRF since going beyond 1 km goes into the "gray zone" or "Terra Incognita" (Wyngaard, 2004) where turbulence starts to be explicitly resolved, and thus not easy to parameterize.*"

Having 100 m grid spacing on the fire grid only affects the elevation and fuel representation. We can see on Figure 4 that the fuels are mostly two fuel classes, category 8, closed timber litter, and no-fuel, and cover large regions well represented by the 100 m spacing. Elevation is more heterogeneous, but at 100 m we will resolve the most important terrain features. So 100 m captures the main heterogeneities over the region. In any case, the important aspect of our experimental design is to have as close as possible configurations in UFS-CFBM and WRF-Fire to check for consistency of the results from both models and this is possible using 100 m grid spacing in the fire grid of both models. Indeed both models have similar elevations and the fuel types since both models use Geogrid to initialize the fire grid. We now say: "*Hence, both UFS and WRF use a fire domain of 100 m grid spacing which is sufficient to capture the main heterogeneities in fuels and elevation over the region. The fuel and elevation data are generated by Geogrid in both models which is desirable for our main purpose of checking consistency between both models.*"

COMMENT 6

The rationale behind different configurations between WRF and UFS regarding vertical layers and time steps should be justified. For instance, the use of different number of vertical layers and time steps?

RESPONSE

The reviewer is right. In the new version of the manuscript we use the same number of vertical levels (65) and the same time step (4 s) for both the atmosphere and the fire model in order to have WRF and UFS configured as close as possible. For the same reason, we also use the same set of parameterizations. We introduced changes to explain these aspects in the new version of the manuscript (see changes in Section 4.1.).

COMMENT 7

*Validation:*
*Adjusting wind height to better match fire progression raises concerns about the validity of the model's predictive capability. It should be clarified whether this adjustment is an empirical correction or an inherent part of the modeling approach.*

RESPONSE

In the new version of the manuscript we do not need to adjust the height of the winds in UFS. The differences were originated by the treatment of the land use in WRF and UFS that caused a discrepancy in the roughness length which largely determines the magnitude of near surface winds. We now use the same land surface model and the roughness lengths are in better agreement between WRF and UFS. This leads to better agreement of the simulated winds. The correlation, bias, and mean absolute error between the wind speed from UFS and WRF is 0.69, -0.3 m/s, and 0.5 m/s, respectively. This is a decent agreement considering UFS and WRF use different dynamical cores. See the new Figures 7 and 9 and their related discussion.

COMMENT 8

*The similarity of results between one-way and two-way coupling suggests that the two-way coupling mechanism has not been adequately validated. Further evidence of feedback interactions is needed.*

RESPONSE

The feedback from the fire to the atmosphere in UFS-CFBM and WRF-Fire is small but there is consistency between the two models as exemplified by the time series of the winds (Figure 9) and fluxes (new Figure 10); the heat flux, temperature, and humidity in the atmospheric grid (new Figs. 11-13); and new in the new version of the manuscript, the vertical velocity (Fig. 14) which is further evidence of feedback interactions. WRF-Fire is widely used and it has been extensively validated, and UFS-CFBM produces impacts in the atmospheric grid of similar magnitudes. In the new version of the manuscript we use 1 km grid spacing in the atmospheric grid and we also see similarities between two-way and one-way simulations in both models, WRF and UFS. This suggests that going to even small scales to explicitly resolve turbulence would be necessary to better capture the fire-atmosphere interactions. We recognize that this is desirable but it is not possible in UFS (see answer to comment 5). However, this is fine for our

experimental design that focuses on ensuring consistency between WRF-Fire and UFS-CFBM. To this end, we show the agreement of the time series of the fire fluxes in the fire grid (new Fig. 10) and in the atmospheric grid (Figs. 11-14). These figures show that the feedback from the fire to the atmosphere is consistent between WRF-Fire and UFS-Fire. We show in Figure 11 the magnitude of the heat flux in the atmospheric grid and how this is, as expected, increasing the 2 m temperature (Fig. 12) and the vertical velocity (Fig. 14). Results for the impacts in the vertical velocity have been added to provide further evidence of the feedback interactions. Again both models are in sufficient agreement to suggest that the feedback is well implemented.

COMMENT 9

*The validation would benefit from quantitative evaluation metrics such as the Jaccard index or Sørensen–Dice coefficient to assess model performance systematically.*

RESPONSE

Yes, we agree that it is good to have quantitative metrics and we have added them to the new version of the manuscript. We have calculated the Heidke skill score and the Sorensen coefficient. Both show similar results. In the new version of the manuscript we show results for the Heidke skill score comparing the results from UFS-CFBM and WRF-Fire, as well as CFBM and WRF-Fire (Fig 1 of this document, and Fig. 6 in the new version of the manuscript). We focus on the comparison of the simulated results because in the manuscript we focus on the model-intercomparison to ensure the consistency of our implementation. The Heidke skill score is always larger than 0.8 which is large. We have added text in the new version of the manuscript to introduce the figure and support the visual comparison of results.

[Figure]

**Figure 1**: Heidke skill score comparing the simulated perimeters from UFS to the WRF-Fire results. The simulation labelled as CFBM is the standalone run driven by WRF data and has been compared against WRF-Fire results

COMMENT 10

*While the manuscript acknowledges underestimating fire spread in some regions and overestimating in others, it does not provide sufficient explanation for these discrepancies. A sensitivity analysis on parameter uncertainties (e.g., fuel properties, and wind corrections) would strengthen the discussion.*

RESPONSE

We have redone the simulations using the same set of parameterizations, the same time step, and the same number of vertical layers. We have also quantified the agreement of the simulated fire areas using the Heidke skill score (see Fig. 1 of this document and the answer to comment 9 above). We describe the discrepancies and provide some explanations such as the presence of barriers not accounted for by the model, portions of the fire perimeter does not seem to be active but in the model the whole perimeter is considered active, and potential issues in the timestamps of the fire perimeters. It is not our objective to evaluate the simulation against observations beyond fire perimeters to characterize/improve the realism of the simulation. That is the logical next step. Herein we focus on the model intercomparison to inspect the adequacy of our implementation.

We did run sensitivity experiments as suggested. We performed sensitivity experiments to the height used to drive the fire evolution and to the fuel moisture content. Results are summarized in Figure 2 of this document that shows the evolution of the burnt area. We see the expected sensitivities, a faster propagation when the fuel moisture content is reduced or the height of the winds is increased. The more important aspect to highlight is the consistency between UFS-CFBM and WRF-Fire results.

[Figure]

**Figure 2**: Evaluation of the size of the burnt area as a function of time for several experiments using WRF-Fire and UFS-CFBM. See legend for the description of the experiments.

COMMENT 11

*Other Minor Comments:*
*Line 156: The phrase "substantial portions of new code" requires further clarification regarding the specific novel contributions.*

RESPONSE

WRF-Fire code is mostly confined to the physics directory in Figure 1. And even this code has been largely modified to eliminate atmospheric dependencies and other parts of the code not needed. There are some WRF subroutines in other directories that we modified from WRF, but the majority outside the physics directory is new code. We have clarified this in the new version of the manuscript: *"The WRF-Fire methods are mostly confined to the physics directory (Fig. 1). We have reorganized, modified and added substantial portions of new code (mostly in the directories not labelled as physics in Fig. 1) to create a standalone fire model,..."*

COMMENT 12

*Line 166: The term "atmospheric dependencies" is ambiguous and requires clarification.*

RESPONSE

Yes, we now say: *"In addition, all the code related to WRF's atmosphere has been removed from the fire code in the physics directory. This includes removing procedures to interpolate atmospheric variables from the atmospheric grid into the fire grid."*

COMMENT 13

*Lines 166-167: The phrase "the WRF grid to the WRF-Fire grid" is misleading, as both the atmospheric and fire grids are components of WRF-Fire.*

RESPONSE

Yes, we now say: *"This includes removing procedures to interpolate atmospheric variables from the atmospheric grid into the fire grid."*

COMMENTS 14 and 15

*Lines 167-168: The statement "The calculation of the fire grid latitude and longitudes..." appears misplaced within the section.*

*Lines 168-169: The manuscript does not adequately describe the improved approach for geolocation determination or how it enhances accuracy.*

RESPONSE

We have modified the sentence related to the geolocation of the fire grid to better connect it to the previous one. We have clarified that we use parameters of the map projection to calculate the fire geolocation just as the geolocation of the atmosphere is done: "*In addition, all the code related to WRF's atmosphere has been removed from the fire code in the physics directory. This includes removing procedures to interpolate atmospheric variables from the atmospheric grid into the fire grid. The interpolation procedures were used to interpolate the latitudes and longitudes from the atmosphere into the fire grid, and we now calculate the geolocation of the fire grid using the map projection information just as the atmospheric grid geolocation is calculated.*"

COMMENT 16

*Line 174: The methodology for handling grid mismatches and specifying the fire domain should be explicitly detailed.*

RESPONSE

Yes, this was missing in the previous version of the manuscript. We have clarified this: "*The standalone model does not need the fire grid, specified by the Geogrid output, to match the WRF domain. The only requirements are the fire grid included within the WRF domain, and the WRF simulation to have certain variables available. The interpolation of variables from the WRF atmospheric data into the fire grid uses a nearest-neighbour interpolation. We plan to include other interpolation methods in future versions of the model.*"

COMMENT 17

*Line 176: ... the three-dimensional horizontal wind components (U and V variables in WRF).*

RESPONSE

Yes, this is correct, we have introduced this change.

COMMENT 18

*General Comments:*

*The manuscript presents limited innovations in fire physics, appearing to largely replicate WRF-Fire, with minor modifications following Mandel et al. (2011) and Munoz-Esparza et al. (2018). The novelty of the contribution should be better articulated.*

RESPONSE

The main novelty of our contribution is to develop, for the first time, a fire behavior model that is available for coupling to atmospheric models via the ESMF library. Using ESMF largely simplifies the coupling of the atmosphere to the fire modules. Indeed, the ESMF library is a standard approach used to couple Earth system components. There is no other fire behavior model with this functionality. We highlight this in the abstract: *"with this aim, we have created, for the first time, a fire behavior model that can be connected to other atmospheric models without the need of developing specific low-level procedures for the particular atmospheric model being used."*. We say this in the introduction: *"Herein we present, for the first time, a fire behavior model implemented as a NUOPC to facilitate its coupling with other  atmospheric models"*. We now highlight the originality in the conclusions as well: *"CFBM is the first fire behavior model available for coupling via the ESMF library."* We believe this originality is appropriate for our model description manuscript. And our results show the consistency of WRF-Fire and UFS-CFBM which supports the adequacy of our implementation. WRF-Fire methods will be enhanced in future versions of the model.

Also, UFS did not have a fire behavior model and UFS users are able for the first time to perform coupled atmosphere-fire behavior simulations. We present here, for the first time, fire behavior simulations based on the UFS model. We have highlighted this in the conclusions: *"The CFBM has been coupled to UFS and this allowed us, for the first time, to perform fire behavior simulations with UFS."*

COMMENT 19

*The study is restricted to a single fire event (Cameron Peak Fire), which limits the generalizability of the findings. Expanding the analysis to multiple fire events with varying topographies and meteorological conditions would improve the robustness of the conclusions. If additional events cannot be incorporated, at a minimum, a sensitivity analysis should be conducted.*

RESPONSE

One fire case is sufficient to check for consistency between WRF-Fire and UFS-CFBM, our main purpose here. We do show sensitivities to the point versus perimeter initialization, and to one versus two way feedback. And we simulate different phases of the fire. In all of the runs and sensitivity experiments we see consistency between both models. See the Heidke skill score and related discussion above (Fig. 1, and answer to comment 9).

The new version of the manuscript now includes a new section showing results for idealized fire simulations. This compares the simulation of the fire spread to the theoretical solution, and we found that it behaves as expected. See results and discussion in the new section 4.2.1.

We believe these results are sufficient to support the conclusions of the study focussed on the adequacy of our implementation and the consistency between WRF-Fire and UFS-CFBM.

We have also done sensitivity experiments to the fuel moisture content and to the height of the winds that drive the fire evolution. We also see consistent results between UFS-CFBM and WRF-Fire. See Fig. 2 above and related discussion in comment 10.

COMMENT 20

*The computational trade-offs of coupling CFBM with other atmospheric models should be acknowledged and compared to existing solutions such as WRF-Fire.*

RESPONSE

The main trade-off is the requirement of using the ESMF libraries to couple the atmosphere to the fire. This provides advantages, since the fire grid does not need to match the atmospheric grid. The fire grid could be much smaller reducing the computational cost. Also, it is possible to have any kind of atmospheric grid (e.g., unstructured) and ESMF will automatically perform the interpolation of variables from one grid to the other. We already indicated that ESMF is a requirement and now we have added: "…*the region covered by the fire simulation in CFBM does not need to match the atmospheric domain allowing for using smaller fire domains which reduces the computational requirements of the fire model. Also, it is possible to have any kind of atmospheric grid (e.g., unstructured) and the ESMF libraries will perform the interpolation of variables automatically.*". It also facilitates the maintenance of the code since the code of each component is isolated from the rest. The communication is done via the  NUOPC cap of each component. We have improved our explanations of the NUOPC interoperability in Section 2.

COMMENT 21

*Final Decision:*
*The manuscript is not recommended for publication mainly for the following reasons:*
*1. Misalignment with the journal's scope: The manuscript presents a well-structured refactoring of the WRF-Fire codebase to create a more modular and flexible fire behavior model. While the core fire physics remains largely unchanged, the restructuring enhances model interoperability, making it easier to couple with different atmospheric models using ESMF. This modularity is a valuable contribution to fostering broader collaborations within the fire modeling community. However, for this journal, which emphasizes advancements in geophysical modeling, the paper would benefit from further developments that go beyond code refactoring, such as improvements in fire physics or additional parameterizations. The reviewer*

*suggests submitting this manuscript to another journal with a different scope after resolving the issues in the following point.*

RESPONSE

We would not downplay the novelty of having a fire behavior model available as NUOPC for coupling with other Earth System's components. It has not been done before and thus our work is original. Our contribution is incremental. This is a model description paper and from the aims and scope of the journal website we believe we align well with among other things "*Model description papers are comprehensive descriptions of numerical models which fall within the scope of GMD.*", or "*describe model components and modules, as well as frameworks and utility tools used to build practical modelling systems, such as coupling frameworks or other software toolboxes with a geoscientific application*"; and, again, we are original with our contribution because there is not a fire behavior model available for coupling as a NUOPC component.

COMMENT 22

*2. Issues with validation methodology:*
*a. Discrepancies in model configurations (e.g., different vertical levels and parameterizations).*
*b. Inappropriate scales that likely weaken fire-atmosphere feedback.*
*c. Validation primarily relies on qualitative comparisons, which are insufficient for rigorous model evaluation.*

RESPONSE

We now have very close model configurations including the same number of vertical levels, time step, and the same set of parameterizations which address the discrepancies raised. We have now increased the grid spacing to 1 km which is at the limit of what can be achieved with UFS. It is true that we see small impacts of the fire on the atmosphere, but the impacts are large enough to quantify the consistency between UFS-CFBM and WRF-Fire (Figs 9-14, with a new figure that compares the vertical velocity), the main goal of the manuscript. We have now added quantitative comparisons to go beyond qualitative aspects. For example, we provide the correlation, bias and mean absolute error for the agreement of the simulated wind speed and the simulated fluxes. And we have quantified the agreement between the simulated fire perimeters using the Heidke skill score (see Fig. 1 of this document and related discussion in the answer to comment 9).

COMMENT 23

*A final recommendation is that instead of using a real-world fire case with significantly different configurations and scales between the coupled components, the authors should consider an idealized test case that allows for a more controlled and rigorous evaluation of the coupling framework.*

RESPONSE

We have added a new section (Section 4.2.1) where we focus on an idealized case. The idealized case replicates the experiment we performed in our previous work with WRF-Fire (Munoz-Esparza et al. 2018) and illustrates that the more accurate the numerical method used to resolve the level set equation that tracks the location of the fire font, the more consistency with the theoretical solution. The results we obtained are in good agreement with our previous results with WRF-Fire (Munoz-Esparza et al. 2018) and contribute to illustrate the adequacy of our implementation in CFBM of WRF-Fire methods. Results are summarized in the new Figure 4 and its related discussion.

**References**

Almgren et al., 2023: ERF: Energy Research and Forecasting. Journal of Open Source Software, 8(87), 5202, https://doi.org/10.21105/joss.05202

DeCastro, A. L., T. W. Juliano, B. Kosovic, H. Ebrahimian, J.K. Balch, 2022: A Computationally Efficient Method for Updating Fuel Inputs forWildfire Behavior Models Using Sentinel Imagery and Random Forest Classification. Remote Sens., 14, 1447. https://doi.org/10.3390/rs14061447

Eghdami, M., P. A. Jimenez y Munoz, and A. DeCastro, 2025: Sensitivity to the representation of wind for wildfire rate of spread: Case studies with the Community Fire Behavior model. Fire, 8, 135

Kochanski, A. K., K. Clough, A. Farguell, D.V. Mallia, J. Mandel, and K. Hilburn, 2023: Analysis of methods for assimilating fire perimeters into a coupled fire-atmosphere model. Front. For. Glob. Change 6:1203578. doi: 10.3389/ffgc.2023.1203578

Munoz-Esparza, D., B. Kosovic, P.A. Jimenez, and J.L. Coen, 2018: An accurate fire-spread algorithm in the Weather Research and Forecasting model using the level-set method. Journal of Advances in Modelling Earth Systems, 10, 908-926.

Skamarock, W. C., Klemp, J. B., Duda, M. G., Fowler, L. D., Park, S.-H., & Ringler, T. D., 2012: A multiscale nonhydrostatic atmospheric model using centroidal Voronoi tessellations and C-grid staggering. Monthly Weather Review, 140, 3090–3105.

Turney, F. A., P.E. Saide, P.A. Jimenez Munoz, D. Muñoz-Esparza, E.J. Hyer, D.A. Peterson, et al., 2023: Sensitivity of burned area and fire radiative power predictions to containment efforts, fuel density, and fuel moisture using WRF-fire. Journal of Geophysical Research: Atmospheres,128, e2023JD038873. https://doi.org/10.1029/2023JD038873

Wyngaard, J., 2004: Toward numerical modeling in the "Terra Incognita", J. Atmos. Sci., 22, 1816–1826.